# Plasma proteomic biomarker signature of age predicts health and life span

**Toshiko Tanaka[1]\*, Nathan Basisty[2], Giovanna Fantoni[3], Julián Candia[4], Ann Z Moore[1], Angelique Biancotto[5], Birgit Schilling[2], Stefania Bandinelli[6], Luigi Ferrucci[1]**

[1]Translational Gerontology Branch, National Institute on Aging, NIH, Baltimore, United States; [2]The Buck Institute for Research on Aging, Novato, United States; [3]National Institute on Aging, Intramural Research Program, Clinical Research Core, NIH, Baltimore, United States; [4]Laboratory of Human Carcinogenesis, Center for Cancer Research, National Cancer Institute, NIH, Bethesda, United States; [5]Precision Immunology, Immunology & Inflammation Research Therapeutic Area, Sanofi, Cambridge, United States; [6]Geriatric Unit, Azienda Sanitaria toscana centro, Firenze, Italy

**Abstract** Older age is a strong shared risk factor for many chronic diseases, and there is increasing interest in identifying aging biomarkers. Here, a proteomic analysis of 1301 plasma proteins was conducted in 997 individuals between 21 and 102 years of age. We identified 651 proteins associated with age (506 over-represented, 145 underrepresented with age). Mediation analysis suggested a role for partial *cis*-epigenetic control of protein expression with age. Of the age-associated proteins, 33.5% and 45.3%, were associated with mortality and multimorbidity, respectively. There was enrichment of proteins associated with inflammation and extracellular matrix as well as senescence-associated secretory proteins. A 76-protein proteomic age signature predicted accumulation of chronic diseases and all-cause mortality. These data support the use of proteomic biomarkers to monitor aging trajectories and to identify individuals at higher risk of disease to be targeted for in depth diagnostic procedures and early interventions.

**\*For correspondence:**
tanakato@mail.nih.gov

**Competing interests:** The authors declare that no competing interests exist.

## Introduction

The elderly population is rapidly growing around the world, and the age group 80 years and older is expanding faster than any other demographic group (*Colby and Ortman, 2015*). Advanced age is one of the most powerful predictors shared by many chronic diseases that are highly prevalent in the older population and negatively impact function and quality of life. Understanding mechanisms by which aging increase the risk of chronic morbidity is of utmost importance for the development of new intervention strategies aimed at improving health-span in the general population.

While, at the population level, the overall risk of pathology increases with chronological age, the age when pathology first emerges clinically, the rates of pathology development and the age at death are highly variable between individuals. Hence, two individuals of the same chronological age may lay on very different health trajectories. While the characterization of risk factors has proven useful for the identification of individuals at high risk of developing specific diseases, there is currently no robust method to distinguish between individuals at risk of global deterioration of health status. For example, chronic inflammation is the only biomarker that predicts the development of multimorbidity (*Fabbri et al., 2015*). Thus, the development of biomarkers that can discriminate, at an early stage, individuals on different health trajectories with aging is a critical challenge in aging research.

Over the last few years, data from a large number of molecular analytes have been used to screen biomarkers associated with chronological age, specific chronic diseases and global health in relatively large populations (*Horvath, 2013*; *Tanaka et al., 2018*; *Sun et al., 2018*). Among these molecular analytes, proteins are particularly attractive because they are direct biological effectors, and dynamic changes with aging or disease are complex and manifold. Indeed, previous studies on circulating proteins have already shown great promise as potential biomarker signatures of human diseases (*Tanaka et al., 2018*). Past studies have used various proteomic platforms to identify novel age- and disease-related protein biomarkers in plasma, serum, and cerebral spinal fluid (*Tanaka et al., 2018*; *Sun et al., 2018*; *Menni et al., 2015*; *Baird et al., 2015*; *Di Narzo et al., 2017*). While several promising candidate biomarkers have been reported, reproducibility of the findings and the clinical utility of these biomarkers is still uncertain. Highly robust and clinically useful biomarkers will emerge following repeated studies in genetically and geographically diverse cohorts.

Since aging is a complex process, it is likely that multiple biomarkers are needed to capture the physiological mechanisms of accelerated aging that eventually lead to the multiplicity of adverse outcomes that typically occur in older persons. Perhaps the best example is the estimation of 'epigenetic clock' signatures that were developed by examining DNA methylation across a large number of DNA sites to single out those that in a weighted average reproducibly predict chronological age (*Horvath, 2013*; *Bocklandt et al., 2011*; *Human Microbiome Project Consortium, 2012*; *Weidner et al., 2014*). Positive deviations of the epigenetic clock from chronological age are meant to estimate biological age acceleration and have, in fact, been shown that they predict pathologic age-related traits, such as diabetes, cardiovascular disease, cancer, and premature death (*Fauci et al., 2008*; *Chen et al., 2016a*; *Levine et al., 2015*; *Perna et al., 2016*). Other epigenetic biomarkers, still based on DNA methylation, were developed by tuning them against health-related characteristics, including risk factors of cardiovascular disease, mortality and lifespan (*Levine et al., 2018*; *Lu et al., 2019*). While these tools have clearly demonstrated that the aging process determines predictable, non-stochastic epigenetic changes, the predictive power of epigenetic biomarkers are still too limited for extensive clinical use. In addition, the mechanisms for such age-related changes in DNA methylation and their association with health outcomes is unknown, as it is challenging to draw conclusions related to functional pathways. Additionally, ideal clinically useful biomarkers should be directly measurable in highly accessible and non-invasive biological specimens such as plasma or serum.

Our group recently developed an accurate signature of age using data from 76 plasma proteins in a population of 240 healthy subjects between 22 and 93 years of age. This proteomic signature of age was correlated with age-related clinical parameters such as inflammation and adiposity (*Tanaka et al., 2018*). A clear advantage of protein biomarkers compared to other biomarkers is that they are direct biological effectors, and their identification provides important clues to the underlying biological mechanism involved in their association with age and health outcomes. However, whether this proteomic clock is associated with clinically relevant aging outcomes could not be determined because of limited sample size and because the initial study was restricted to extremely healthy individuals.

To address the limitations of our previous US-based study, here we examined associations between age and abundances of 1301 proteins measured using an aptamer-based method on a representative population sample of 997 participants in the Italy-based InCHIANTI study. We aimed to identify novel age-associated proteins, as well as confirm previously reported age-associated proteins, and to uncover their relationship with age-related outcomes. By using a genetically and geographically distinct cohort, we also aimed to crystallize the most robust biomarkers in diverse populations. We used a Mendelian Randomization (MR) approach to explore genetic evidence of causal relationship between age-associated proteins with key chronic diseases. We also explored DNA methylation as an underlying epigenetic mechanism for age-related changes in protein abundance and show that, for some proteins, methylation can partially explain the observed age associations. We further investigated whether proteomic data can be used to predict chronological age in a geographically distinct cohort using the algorithm from our previous study (*Tanaka et al., 2018*). Moreover, we assessed that the deviation between predicted and observed age may reflect rates of aging and could be used as a measure of accelerated biological aging.

## Results

### Association of protein abundance and chronological age

Plasma proteomic profiling was conducted on 997 individuals (45% men, 55% women) between the ages of 21–98 years (average 66.3 ± 15.4 years) from the baseline visit of the InCHIANTI study (*Table 1*, *Table 1—source data 1*). A global burden of chronic diseases calculated as the number of 15 common chronic diseases that affected the participants at the time of assessment, hereafter referred to as multimorbidity, was progressively higher at older ages. The prevalence of all common diseases except for chronic obstructive pulmonary disease and Parkinson's disease was higher in the older age groups (*Table 1*, *Table 1—source data 1*). Of the 997 participants, 504 (50.6%) died over the 18 year follow-up period.

The association of 1301 SOMAmers with chronological age was examined using a linear model. Of the 1301 proteins tested, 651 proteins (506 overrepresented and 145 underrepresented with age) were associated with chronological age at Benjamini-Hochberg false discovery rate (B-H FDR)$\leq$ 0.05 (*Figure 1A*; *Figure 1—source data 1*). The percent variation in protein abundances explained by age ranged from 0.3% to 50.6%. When the analysis was adjusted for the burden of chronic disease, 537 proteins (82.5%) remained significantly associated with age, with small effects on the strength of the association. Specifically, the effect size (or beta estimates) of the regression model changed by 30% or more in only 35 of these proteins (6.5%). These findings suggest that the majority of variance in these proteins was accounted by age 'per se' rather than age-related differences of health status. To validate the finding, we used data from a cohort 240 healthy subjects from the

**Table 1.** Clinical and demographic characteristics of 997 InCHIANTI subjects at baseline visit.

| | Overall | | 20–60 years | | 60–70 years | | 70–80 years | | 80+ years | | p |
|---|---|---|---|---|---|---|---|---|---|---|---|
| n | 997 | | 203 | | 285 | | 380 | | 129 | | |
| Age (yrs) | 66.3 | (15.4) | 40.1 | (11.1) | 66.4 | (2.2) | 73.9 | (2.7) | 84.9 | (3.6) | <0.001 |
| % Women | 549 | (55.1) | 106 | (52.2) | 152 | (53.3) | 207 | (54.5) | 84 | (65.1) | 0.097 |
| % Ripoli | 517 | (51.9) | 98 | (48.3) | 147 | (51.6) | 192 | (50.5) | 80 | (62.0) | 0.084 |
| % death | 504 | (50.6) | 9 | (4.4) | 109 | (38.2) | 259 | (68.2) | 127 | (98.4) | <0.001 |
| Follow-up time (yrs) | 15.0 | (4.9) | 18.4 | (1.6) | 16.6 | (3.8) | 14.1 | (4.8) | 8.7 | (4.0) | <0.001 |
| Number of common diseases | 1.3 | (1.3) | 0.3 | (0.5) | 1.2 | (1.1) | 1.7 | (1.3) | 2.3 | (1.5) | <0.001 |
| Hypertension | 375 | (37.6) | 18 | (8.9) | 120 | (42.1) | 168 | (44.2) | 69 | (53.5) | <0.001 |
| Depression | 183 | (18.4) | 22 | (10.8) | 36 | (12.6) | 88 | (23.2) | 37 | (28.7) | <0.001 |
| Cognitive impairment | 177 | (17.8) | 3 | (1.5) | 29 | (10.2) | 82 | (21.6) | 63 | (48.8) | <0.001 |
| Diabetes | 103 | (10.3) | 4 | (2.0) | 32 | (11.2) | 47 | (12.4) | 20 | (15.5) | <0.001 |
| Lower extremities joint disease | 84 | (8.4) | 1 | (0.5) | 18 | (6.3) | 49 | (12.9) | 16 | (12.4) | <0.001 |
| Peripheral artery disease | 76 | (7.6) | 0 | (0) | 14 | (4.9) | 39 | (10.3) | 23 | (17.8) | <0.001 |
| Anemia | 74 | (7.4) | 10 | (4.9) | 13 | (4.6) | 28 | (7.4) | 23 | (17.8) | <0.001 |
| Ischemic heart disease | 54 | (5.4) | 0 | (0) | 16 | (5.6) | 27 | (7.1) | 11 | (8.5) | 0.001 |
| Chronic obstructive pulmonary disease | 50 | (5.0) | 3 | (1.5) | 18 | (6.3) | 23 | (6.1) | 6 | (4.7) | 0.065 |
| Cancer | 48 | (4.8) | 1 | (0.5) | 18 | (6.3) | 24 | (6.3) | 5 | (3.9) | 0.008 |
| Stroke | 34 | (3.4) | 1 | (0.5) | 3 | (1.1) | 22 | (5.8) | 8 | (6.2) | <0.001 |
| Congestive heart failure | 28 | (2.8) | 0 | (0) | 6 | (2.1) | 16 | (4.2) | 6 | (4.7) | 0.013 |
| Hip fracture | 22 | (2.2) | 0 | (0) | 3 | (1.1) | 12 | (3.2) | 7 | (5.4) | 0.003 |
| Chronic kidney disease | 8 | (0.8) | 0 | (0) | 0 | (0) | 2 | (0.5) | 6 | (4.7) | <0.001 |
| Parkinson's disease | 6 | (0.6) | 0 | (0) | 2 | (0.7) | 2 | (0.5) | 2 | (1.6) | 0.354 |

Data represent mean (SD) for continuous variables and n (%) for categorical variable.

The online version of this article includes the following source data for Table 1:

**Source data 1.** Phenotypic data of the InCHIANTI study.

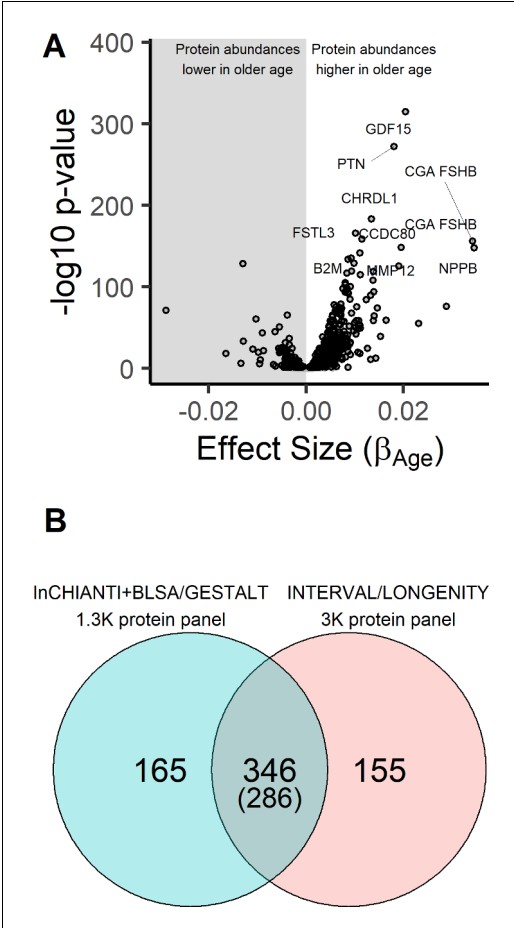

**Figure 1.** Age-associated proteins in the InCHIANTI study. (**A**) The volcano plot displays the results from the association of 1301 proteins with chronological age in the InCHIANTI study (N = 997; *Figure 1—source data 1*). The figure displays the effect size, or the beta coefficient for age (βage) from the linear model, against significance presented as the -log10(P-value). Of the 1301 proteins, 651 proteins associated were associated with age. (**B**) Comparison of the age-associated proteins discovered using the 1.3K SOMAscan platform in the INCHIANTI/BLSA/GESTALT studies (*Figure 1—source data 2*) with the analysis using the 3K SOMAscan platform in the INTERVAL/LONGEVITY study (*Lehallier et al., 2019*). There were 346 proteins that were significant in both studies, and 286 of these proteins the direction of association with age were concordant.

The online version of this article includes the following source data and figure supplement(s) for figure 1:

**Source data 1.** Results from the association analyses of 1301 proteins with chronological age in the InCHIANTI study.

**Source data 2.** Results from the meta-analyses of protein associations with age in the InCHIANTI and BLSA/GESTALT study.

*Figure 1 continued on next page*

Baltimore Longitudinal Study on Aging (BLSA) and the Genetic and Epigenetic Signatures of Translational Aging Laboratory Testing (GESTALT) using in the previous publication (*Tanaka et al., 2018*). Of the 651 proteins that were significantly associated with age in the InCHIANTI study, 60% (395 proteins) were significantly associated with age in the BLSA/GESTALT study. To identify additional age-associated proteins, a meta-analysis of associations results from the BLSA-GESTALT and the InCHIANTI study was performed. There were 735 proteins associated with age 596 higher abundance with age, 139 lower abundance with age; (*Figure 1—figure supplement 1*, *Figure 1—source data 2*). We compared these findings with results from the INTERVAL/LonGenity study using the 3K SOMAscan platform, with 1.3K SOMAscan platform used in the InCHIANTI study. Of the 735 age-associated proteins, 511 were measured in the INTERVAL/LonGenity study and 346 (68%) proteins were significantly associated with age, and more importantly for 286 (56%) proteins, the directions of age association were concordant (*Figure 1B*). There were 224 novel age-associated proteins discovered in the 1.3K platform.

Next, we examined whether association between protein abundance with age was modified by sex. Of the 1301 proteins tested, there were 427 proteins with sex differences in protein abundance, 328 of which were overrepresented in women and 99 overrepresented in men (*Figure 2A*, *Figure 2—source data 1*). As expected, the most significant differences were observed for proteins with known sex differences such as PSA (higher in men), and FSH, leptin, HCG (higher in women) which can be considered as powerful positive controls. To test the robustness of these association, these results were compared to results from INTERVAL study (*Lehallier et al., 2019*). Of the 427 proteins associated with sex in the InCHIANTI study, 294 (68.9%) were measured in the INTERVAL study (*Figure 2—source data 1*). Of these, 230 proteins (78.2%) were associated with sex in the same direction with the INTERVAL study, reflecting the robustness of sex-associated proteins. Next, we tested whether sex modified any of the associations between age and protein abundances. Overall, we discovered 50 proteins that showed significant differences in the association with age-by-sex (*Figure 2B*, *Figure 2—source data 1*). The most significant sex-difference was observed for sex hormone-binding globulin, that showed a strong significant rise with aging in men but no age-trend in women (*Figure 2C*).

*Figure 1 continued*

**Figure supplement 1.** Results from the meta-analysis of age-associated proteins in the InCHIANTI and BLSA/GESTALT studies.

## Age–protein association and mediation by methylation

Several lines of research suggest that DNA methylation affects gene transcription and by this mechanism may also affect protein expression. Given the robust literature indicating that DNA methylation changes systematically with aging, we tested whether the association of protein abundance and age can be explained in part by age-related changes in DNA methylation in circulating white blood cells. In 460 subjects with both methylation and proteomic data, a mediation analysis was conducted to test the hypothesis that changes in protein concentrations with aging are mediated by epigenetic regulation, namely changes in percent methylation in CpG sites located within 10 kb of the protein coding gene. This was performed in four steps. In the first step, we identified proteins that were associated with age in the subcohort with both proteomic and methylation data. Of the 651 age-associated proteins in the full cohort, a subset of 499 proteins were significantly associated with age in this sub-cohort. Of these 499 age-associated proteins, 485 were expressed by genes on autosomal chromosomes. Since some of the aptamer probes target protein complexes, those 485 proteins were encoded by 494 genes. In the second step, we identify age-associated CpG loci. Of the 472,138 CpG sites measured, 170,780 were significantly associated with age. There was significant enrichment of age-associated CpG loci within 10 kb of genes coding for age-associated proteins compared with genes coding for other proteins assayed by the SOMAscan assay used here (41.8% vs 37.9% respectively, p=0.0001). In the third step, the existence of a possible mediation effect was tested by regressing protein abundance with age, adjusted for age-associated CpG. For 449 out of the 494 genes, there was at least one CpG that mediated the association between age and protein abundance ($P_{sobel}$ <0.05; *Supplementary file 1A*). On average 7.8% (range 0.3–31.2%) of the age-associated CpG methylation mediated the association between age and protein abundance. The percent mediation ranged from 2% to 100%. The most significant mediation was observed for ectonucleotide pyrophosphatase/phosphodiesterase 7 (ENPP7), member of the ectonucleotide pyrophosphatase/phosphodiesterase family implicated in phospholipids and cholesterol metabolism with a CpG cg15739835. There is a positive association of ENPP7 abundance with age, and this association is attenuated

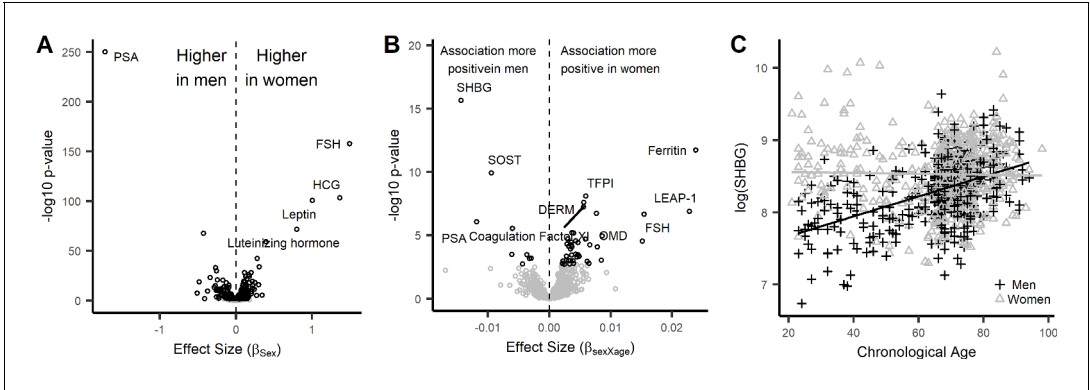

**Figure 2.** Sex-associated proteins and sex differences in age association of plasma proteins in the InCHIANTI study. (**A**) Volcano plot displaying the results from the association of 1301 proteins with sex in the InCHIANTI study (*Figure 2—source data 1*). There were 427 proteins (black circles) with differential expression by sex. The figure displays the effect size, or the beta coefficient for sex (βage) where a positive value are proteins with overrepresentation in women, and a negative value are proteins overrepresented in men. The y-axis show the significance presented as the -log10 (P-value). (**B**) Volcano plot displaying the analysis to explore the differences in age association by sex (*Figure 2—source data 1*). The values represent the beta estimates for the interaction term between sex and age from the linear model. There were 50 proteins (black circles) with differential age association by sex. A positive effect size are proteins where the age association is more positive in women, conversely a negative effect size reflect proteins where the age association is more positive in men. (**C**) Scatterplot displaying the relationship between protein sex hormone binding globulin (SHBG) and age stratified by sex. In men, SHBG values were higher in older ages (+) whereas no age association was observed in women (Δ).

The online version of this article includes the following source data for figure 2:

**Source data 1.** Results from the association analyses of protein with sex and interactions between sex and age.

when adjusted for cg15739835 methylation (*Figure 3A*). There is negative association between cg15739835 methylation with age, reflecting lower methylation at older age at this CpG site (*Figure 3B*). These data suggest that the higher abundance in ENPP7 at older age may be explained by lower gene silencing since methylation is reduced at older age.

## Association of age-related proteins with all-cause mortality

The association of age-associated proteins with all-cause mortality was tested with cox proportional hazards model. Of the 651 age-associated proteins in the InCHIANTI study, the model for 26 proteins violated the proportional hazards assumption. Of the remaining 625 proteins, 497 proteins were associated with all-cause mortality and 193 of them remained significantly associated with mortality after adjusting for covariates (age, sex, study site) (*Figure 4—source data 1*). For the 26 proteins where the proportional hazard assumption was not met, stratified analyses showed that 25 of the proteins were predictive of all-cause mortality in the first time interval of 10 or 15 years (*Figure 4—source data 2*). Among the proteins that were associated with age, there was an enrichment for inflammatory pathways (TNF-activated receptor activity and chemokine receptor binding), regulation of gene expression (DNA methylation, meiosis, epigenetic regulation of gene expression), and extracellular matrix (activation of matrix metalloproteinases, basement membrane, extracellular matrix organization) (*Figure 4A*). The most frequent pathway annotations among proteins were interleukin-10 signaling (35.71%), EPH receptor signaling (28.57%), and chemokines (7.14%). Cellular senescence is widely regarded as a basic aging process that drives numerous pathologies of aging via the secretion of a protein milieu known as the senescence-associated secretory phenotype (SASP) (*Coppé et al., 2008*). Senescent cells and the SASP are a potential source of circulating pro-aging factors in plasma (*Wiley et al., 2019*). We previously reported an enrichment of core SASP factors among plasma biomarkers of healthy aging (*Tanaka et al., 2018*; *Basisty et al., 2020*). Among 175 mortality-associated proteins that increased with age, we identified 13 core SASP factors, including 3 of the four top core SASP signature proteins recently described (*Basisty et al., 2020*) - GDF15, MMP1, and STC1 – and other extensively reported classical SASP factors (IGFBP2 and 4, TIMP1 and 2, and IL-6) (*Figure 4B*). Chronological age alone was a strong predictor of all-cause mortality (C-index 0.78). The plasma proteins most predictive of all-cause mortality in a univariate model were GDF15 C-index 0.75IGFBP2 (C-index 0.73), and B2M (C-index 0.72). After adjustment for age, sex and study site (C-index 0.79), the most predictive proteins were GDF15 (C-index 0.80), TFF3 (C-index 0.80), and PI3 (C-index 0.80; *Figure 4C*). In a multi-protein model, the best

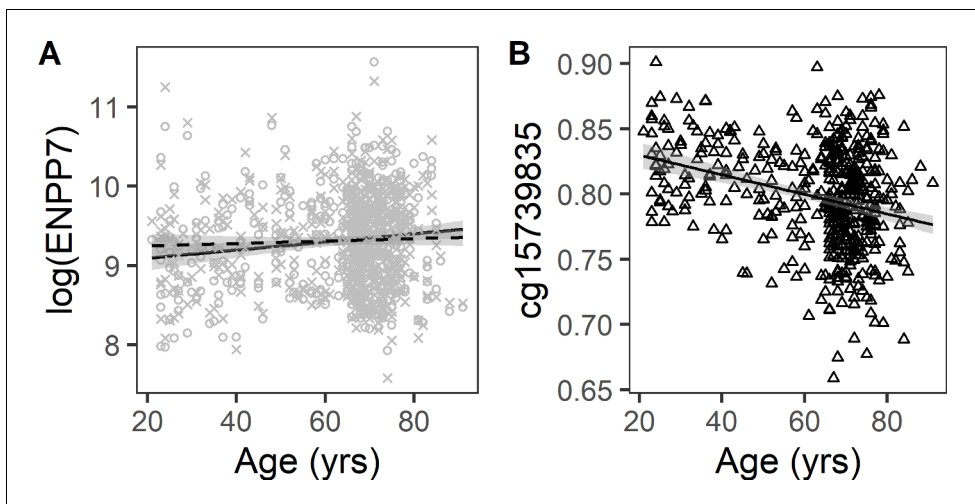

**Figure 3.** Association of plasma ENPP7 with age and mediation by cg15739835. Mediation analysis in 460 subjects with DNA methylation and proteomic data was conducted with Baron and Kenny method (*Supplementary file 1A*). The most significant mediation was observed for ectonucleotide pyrophosphatase/phosphodiesterase 7 (ENPP7). (**A**) Scatterplot displaying the association of plasma ENPP7 with age (solid line) and attenuation of the association after adjustment for methylation at cg15739835. (**B**) Scatterplot displaying the negative association between methylation at cg15739835 with age.

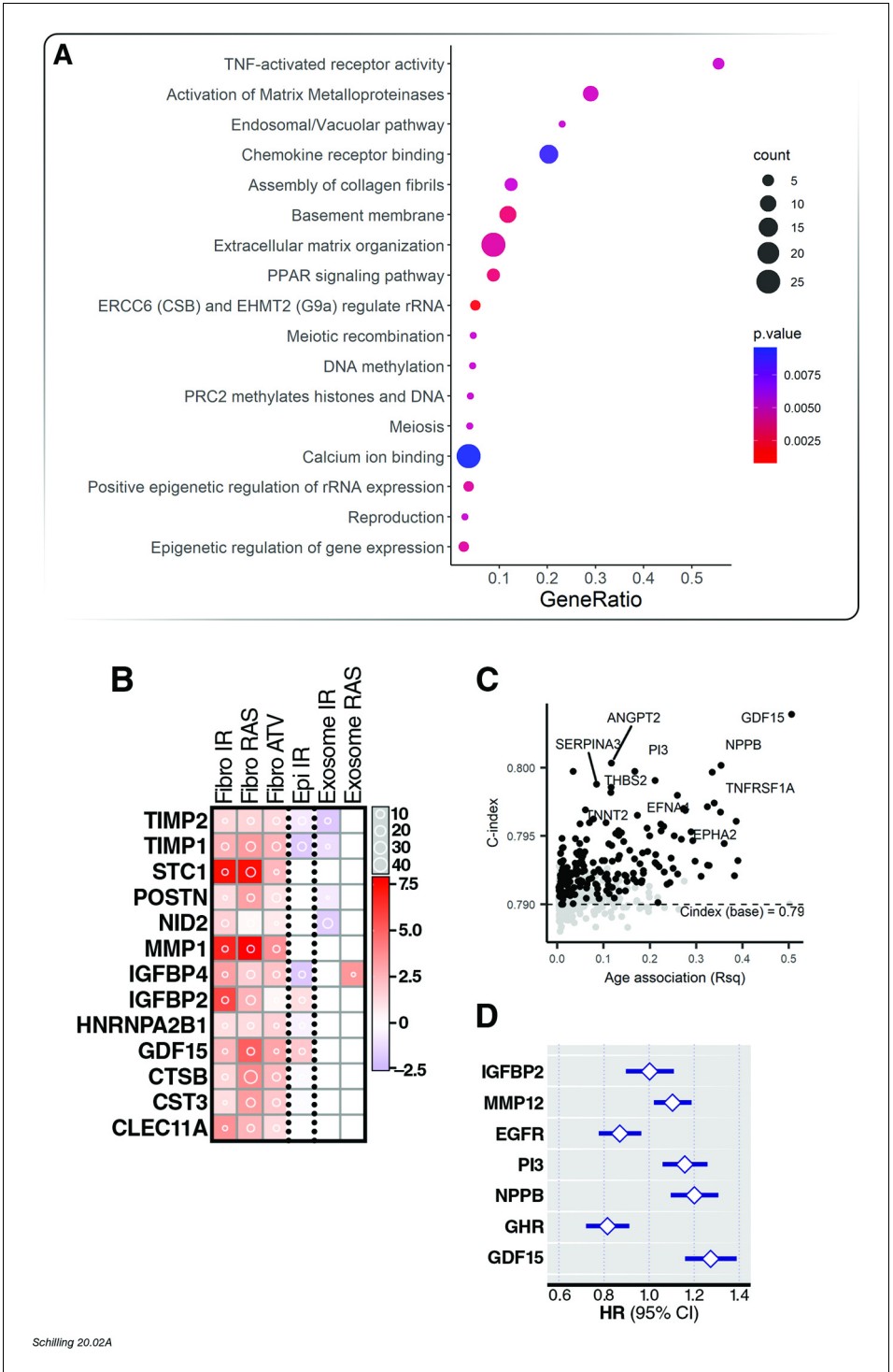

**Figure 4.** Association of age proteins with all-cause mortality. Of the 651 age-associated proteins, 218 was associated with all-cause mortality (*Figure 4—source data 1*). (**A**) Pathway enrichment analysis of age-associated proteins predictive of all-cause mortality. (**B**) There were 13 core senescence-associated secretory proteins (SASP) among the proteins that predict mortality. (**C**) Scatterplot displaying the performance (c-index) of plasma proteins that predict all-cause mortality (y-axis) against the proportion of variance in age that is explained by the biomarker (Rsq) on the x-axis. (**D**) Forest plot to display the effect of 8 plasma proteins that significantly predicted all-cause mortality in a multi-protein model.

The online version of this article includes the following source data for figure 4:

**Source data 1.** Results from association of age-associated proteins with mortality in the InCHIANTI study.

**Source data 2.** Results from association of age-associated proteins with mortality considering interactions with time interval in the InCHIANTI study.

---

model is composed of 8 predictors, which includes age and proteins IGFBP2, MMP12, EGFR, NPPB, GDF15, PI3, and GHR (C-index 0.84, *Figure 4D*).

## Association of age-related proteins with multimorbidity

We next tested the association of age-related proteins with prospective development of 15 common diseases (see Materials and methods). On average, there was an increase in 0.18 comorbid diseases per year over a 10 year follow-up period in the study cohort. Of the 651 age-associated proteins, 549 proteins were associated with changes in multimorbidity, and 295 proteins remained significantly associated with subsequent rates of change in multimorbidity after adjustment for covariates (baseline age, sex, study site) (*Figure 5—source data 1*). Morbidity-associated proteins were strongly enriched for inflammatory pathways (interleukin-7 signaling, cytokine receptor interaction, senescence-associated secretory phenotype), regulation of IGF transport and uptake by IGFBPs, regulation of gene expression (HDMs demethylate histones, meiosis, SIRT1 negatively regulates rRNA, histone modifications, DNA methylation), and amyloid fiber formation (*Figure 5A*). However, the most frequent pathway annotations were cell proliferation (42.4%), receptor regulator activity (20.3%), and chemotaxis (10.2%). Among 247 multimorbidity-associated proteins that increased with age, we identified 17 core SASP factors, including three top core SASP proteins - GDF15, MMP1, and STC1 – and other extensively reported classical SASP factors (IGFBP2, 4, 5, and 7, TIMP1 and 2) (*Figure 5B*). Among the SASP, 11 proteins were associated both with mortality and multimorbidity.

As expected, older chronological age was significantly associated with greater increase in the number of comorbid diseases (β = 0.005 (0.002), p<0.001). The proteins most significantly associated with prospective rise of multimorbidity were GDF15 (b = 0.072 [0.003], B-H FDR = $8.64 \times 10^{-88}$), NPPB (b = 0.066 [0.003], B-H FDR = $9.27 \times 10^{-71}$), and PTN (b = 0.064 [0.003], p=$5.01 \times 10^{-65}$). After adjustment for age, sex and study site, the proteins most significantly associated with prospective rise of multimorbidity were GDF15 (b = 0.044 [0.04], B-H FDR = $4.370.53 \times 10^{-19}$), NPPB (b = 0.038 [0.004], B-H FDR = $2.78 \times 10^{-18}$), and TFF3 (b = 0.033 [0.004], B-H FDR = $4.90 \times 10^{-17}$); *Figure 5C*. The most predictive multi-protein model included age and proteins NPPB, GDF15, MMP12, SMOC1, TNFRSF1B, and FSTL3 (Adj-Rsq = 0.43, *Figure 5D*).

We further investigated whether age-associated proteins can predict development of multimorbidity in subjects who were free of disease at baseline (N = 310). On average, there was an increase in 0.13 comorbid diseases per year over a 10 year follow-up period in these participants. Of the 651 age-associated proteins, 285 proteins were associated with increase in multimorbidity, and 20 proteins were associated independent of age, sex and study site (*Figure 5—source data 1*). The proteins most significantly associated with prospective rise of multimorbidity were PTN (b = 0.052 [0.01], B-H FDR = 0.004), TNFRSF1A (b = 0.038 [0.004], B-H FDR = 0.004), and ANGPT2 (b = 0.036 [0.009], B-H FDR = 0.007); *Figure 5E*. The most predictive multi-protein model included age and proteins PTN, MMP12, CHI3L1, FABP3, and CCDC80 (Adj-Rsq = 0.45, *Figure 5F*).

## Differential protein expression across age-span

Lehallier et al. described important non-linear expression of proteins with age, with distinct waves of differential expression at the fourth, sixth and eight decade of life (*Lehallier et al., 2019*). Using the sliding-window analysis developed by the group, we explored whether similar patterns of proteomic changes could be observed in our data. Consistent with the results from the study by Lehallier and colleagues (*Lehallier et al., 2019*), there was a peak wave of differential expression at 78 years of age where 635 proteins were differentially expressed using a 20 year age bin (*Supplementary file 1B*). These findings were confirmed in analyses that used alternative age bins of 10 and 30 years (*Figure 6A*). There were no clear peaks at younger decades. Of the proteins identified at older age (at 78 years), 63% had been already captured using a linear model. To explore the clinical significance of the proteins identified using linear models in the sliding-window analysis, we tested whether there was enrichment of proteins associated with age-related diseases in the ranked list of age-associated proteins at 78 years, 61 years, and 41 years as well as age-associated proteins from

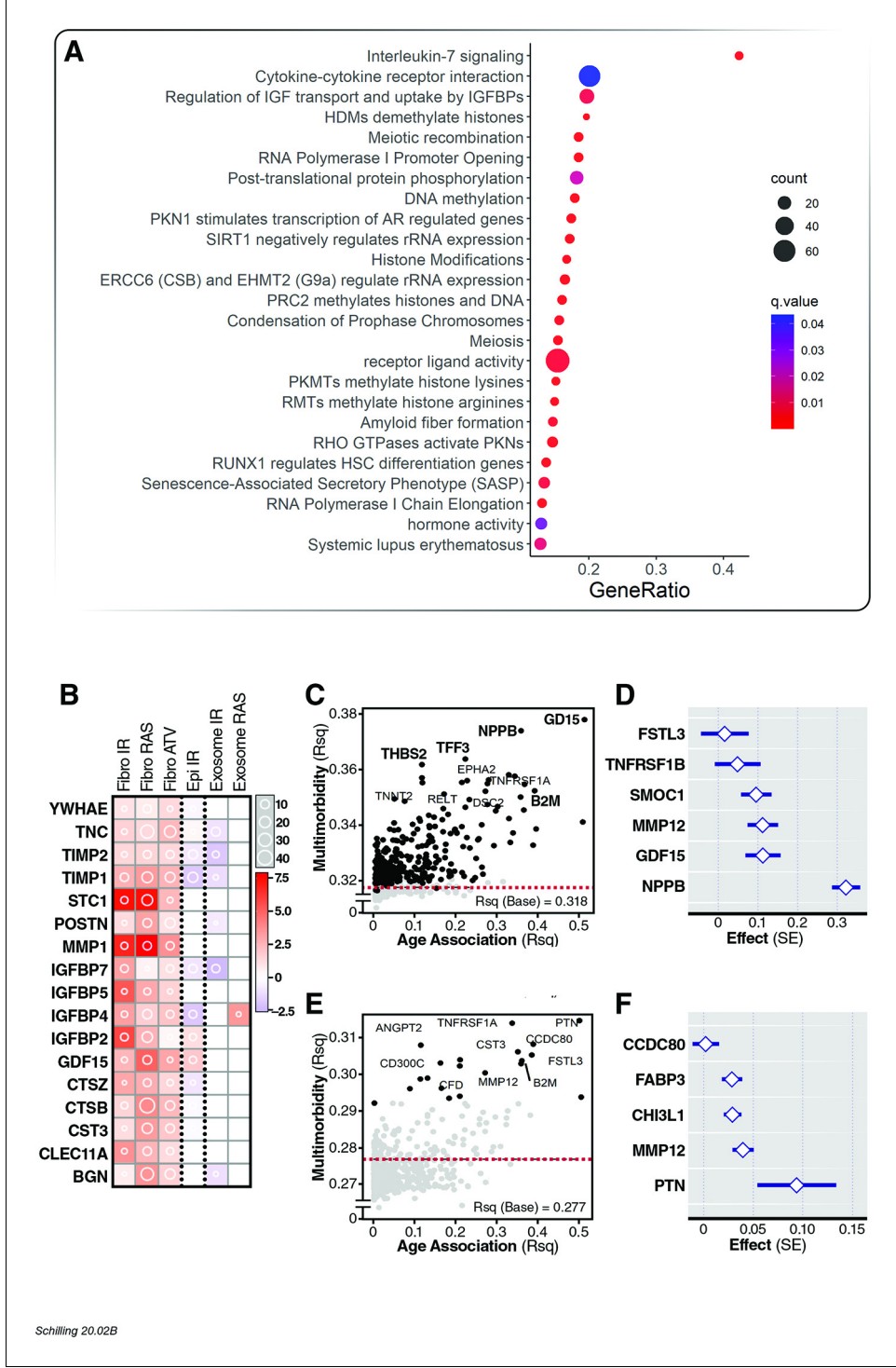

**Figure 5.** Association of age proteins with multimorbidity. Of the 651 age-associated proteins, 295 was associated with all-cause mortality (*Figure 5—source data 1*). (A) Pathway enrichment analysis of age-associated proteins predictive of multimorbidity. (B) There were 13 core senescence-associated secretory proteins (SASP) among the proteins that predict mortality. Scatterplot displaying the proportion of variance (Rsq) of age explained by protein (x-axis) with relationship (Rsq) with protein with baseline multimorbidity (y-axis) (C) and trajectory of multimorbidity over 10 years in subjects free of disease at baseline (y-axis) (E). Forest plot to display the effect of six and five plasma proteins that predict baseline (D) and trajectory of multimorbidity (F).

The online version of this article includes the following source data for figure 5:

**Source data 1.** Results from association of age-associated proteins with number of diseases in the InCHIANTI study.

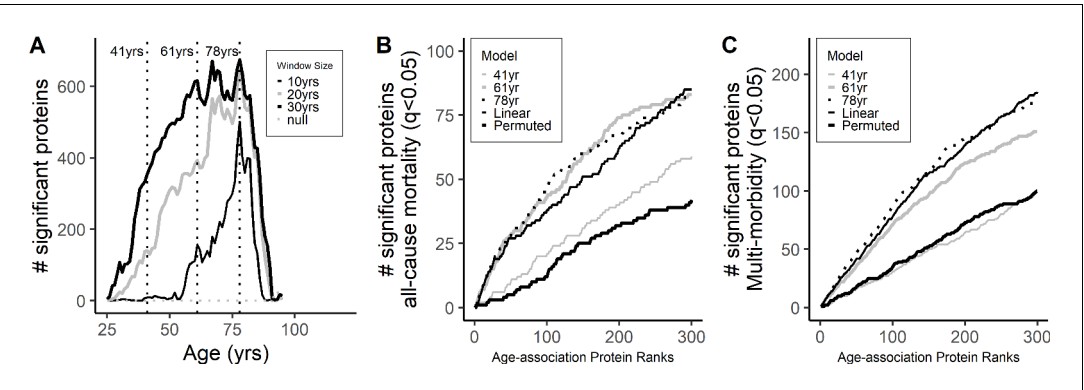

**Figure 6.** Differential protein expression across age-span. (A) Differential expression of protein across different ages was explored suing a sliding-window analysis using an age window of 10, 20, and 30 years (**Supplementary file 1B**). Peak association was observed at age 78 while no clear peaks were observed at younger ages. Cumulative number of proteins associated with all-cause mortality (A) and multimorbidity (B) was compared in the ranked age-associated proteins at the different age waves (41, 61 and 78 years) and from the linear models.

the linear model. There was significant enrichment of proteins associated with all-cause mortality (**Figure 6B**) and multimorbidity (**Figure 6C**) in proteins identified in the linear model and at 61 years and 78 years but not at 41 years.

## Mendelian randomization

To explore causal relationship between genetically determined plasma proteins and age-related phenotypes, we implemented a MR strategy using genetic variants from previously published protein quantitative trait loci (pQTL) analyses of plasma proteins for 10 proteins (GDF15, MMP12, NPPB, GHR, PI3, SMOC1, TNFRSF1B, IGFBP2, EGFR, FSTL3) identified across analyses of multimorbidity and mortality (**Sun et al., 2018**). Aging traits that reflect the diseases in the multimorbidity index were targeted as outcomes, including hypertension, type 2 diabetes, ischemic heart disease, myocardial infarction, ischemic stroke, cardioembolic stroke, prostate cancer, lung cancer, ovarian cancer, Parkinson's disease, depressive disorder, Alzheimer's disease, and rheumatoid arthritis as binary variables, as well as bone mineral density, estimated bone mineral density (BMD), hemoglobin and blood cell distribution width as continuous traits (**Supplementary file 1C**). We found that genetically higher NPPB was associated with lower risk of hypertension (OR = 0.974 [0.947, 0.981]); MMP12 with lower risk of CHD (OR = 0.94 [0.91, 0.98]), MI (OR = 0.94 [0.90, 0.98]), and ischemic stroke (OR = 0.90 [0.84, 0.96]); PI3 with increased risk of MI (OR = 1.07 [1.00, 1.15]), both ischemic (OR = 1.12 [1.01, 1.25]) and cardioembolic stroke (OR = 1.26 [1.02, 1.55]), lung cancer (OR = 1.20 [1.06, 1.36]), and rheumatoid arthritis (OR = 1.11 [1.01, 1.23]); SMOC1 with higher eBMD (β = 0.03 [0.01, 0.06]), Hb (β = 0.05 [0.02, 0.07]), lower RDW (β = −0.10 [-0.13,–0.08]); and GHR with lower eBMD (β = −0.05 [-0.09,–0.01]) (**Figure 7A,B**). No significant results were observed with genetic instruments for GDF15 and TNFRSF1B.

## Proteomic signatures of age

We show that individual age-related proteins are predictive of both all-cause mortality and accumulation of multimorbidity independent of chronological age. We previously developed proteomic signature of chronological age based on 76 proteins (**Supplementary file 1D**; **Tanaka et al., 2018**). Recent data suggest that the proteomic clock developed based on the aptamer-based technology is robust. Interestingly, Lehallier et al developed a parsimonious proteomic clock based on nine proteins using data from the ~3000 protein SOMAscan platform (**Lehallier et al., 2019**). Of the nine proteins in this proteomic clock, six proteins were measured by the 1.3K platform used in the InCHIANTI study, and five out of the six proteins measured are part of the 76-protein signature. There is high correlation between the predicted age based on the two proteomic clocks (r = 0.88; **Figure 8—figure supplement 1**) as well as the measure of age acceleration based on the two clocks (r = 0.65; **Figure 8—figure supplement 1**).

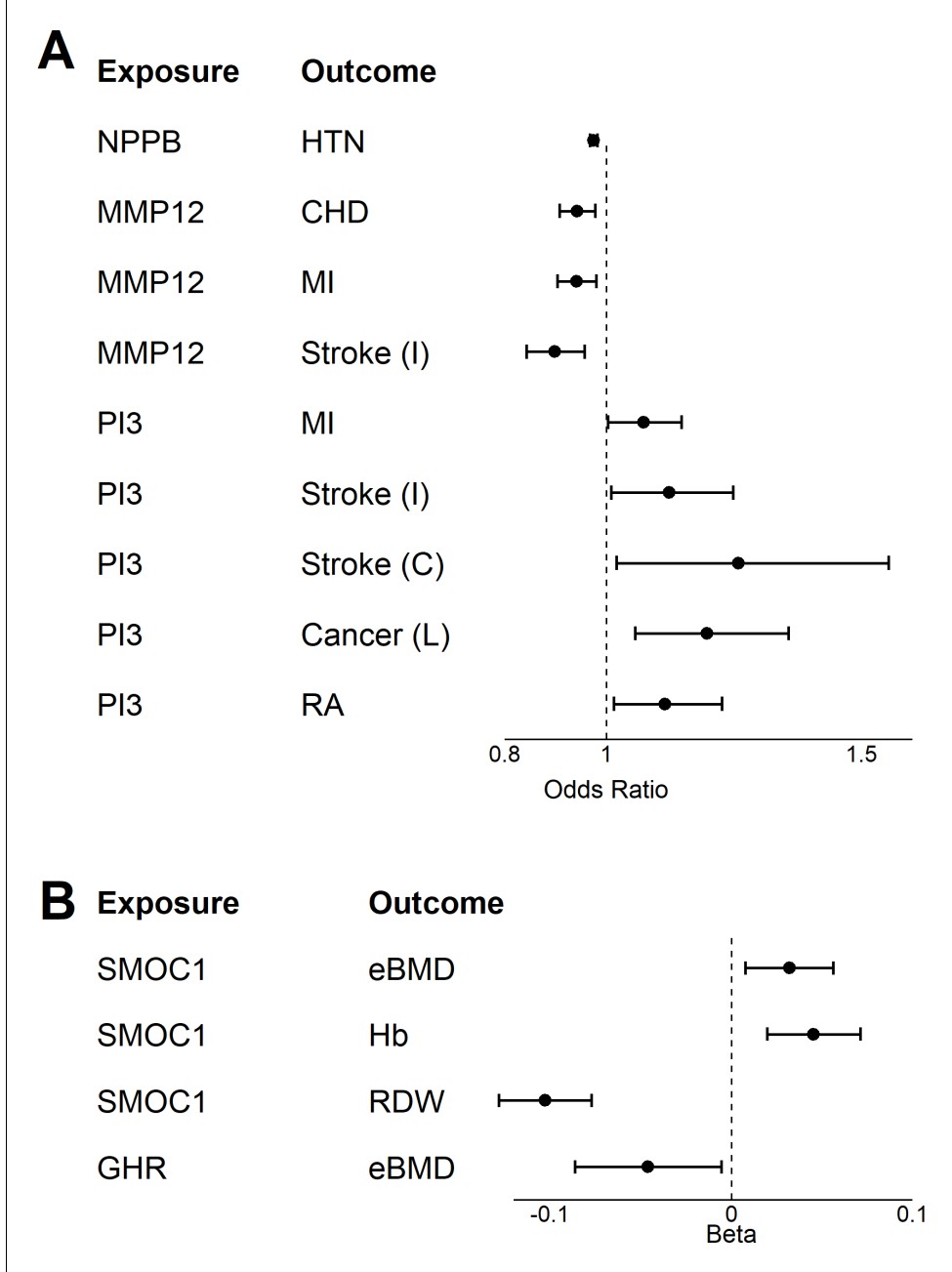

**Figure 7.** Two-sample Mendelian Randomization. A two-sample Mendelian randomization (MR) was conducted for 18 age-related disease and endophenotypes using genetic instrument identified in the INTERVAL study for seven age-associated proteins (*Sun et al., 2018*) and (*Supplementary file 1C*). Forest plot displaying the results from the two-way MR for age-related disease (**A**) and endophenotypes (**B**).

We tested whether the 76-protein signature of age is associated with all-cause mortality and multimorbidity in the InCHIANTI study. The correlation between predicted age (*PROage*) from the 76-protein signature and observed chronological age in the InCHIANTI study was 0.87 (*Figure 8A*). Since the BLSA/GESTALT cohort included subjects who were free of physical and cognitive diseases, we applied the same health criteria to the InCHIANTI study. When examining just within the subjects that met the 'healthy' criteria, the correlation between *PROage* and observed age was 0.91. In the subjects who did not meet the 'healthy' criteria, the correlation was lower at 0.76 (*Figure 8A*).

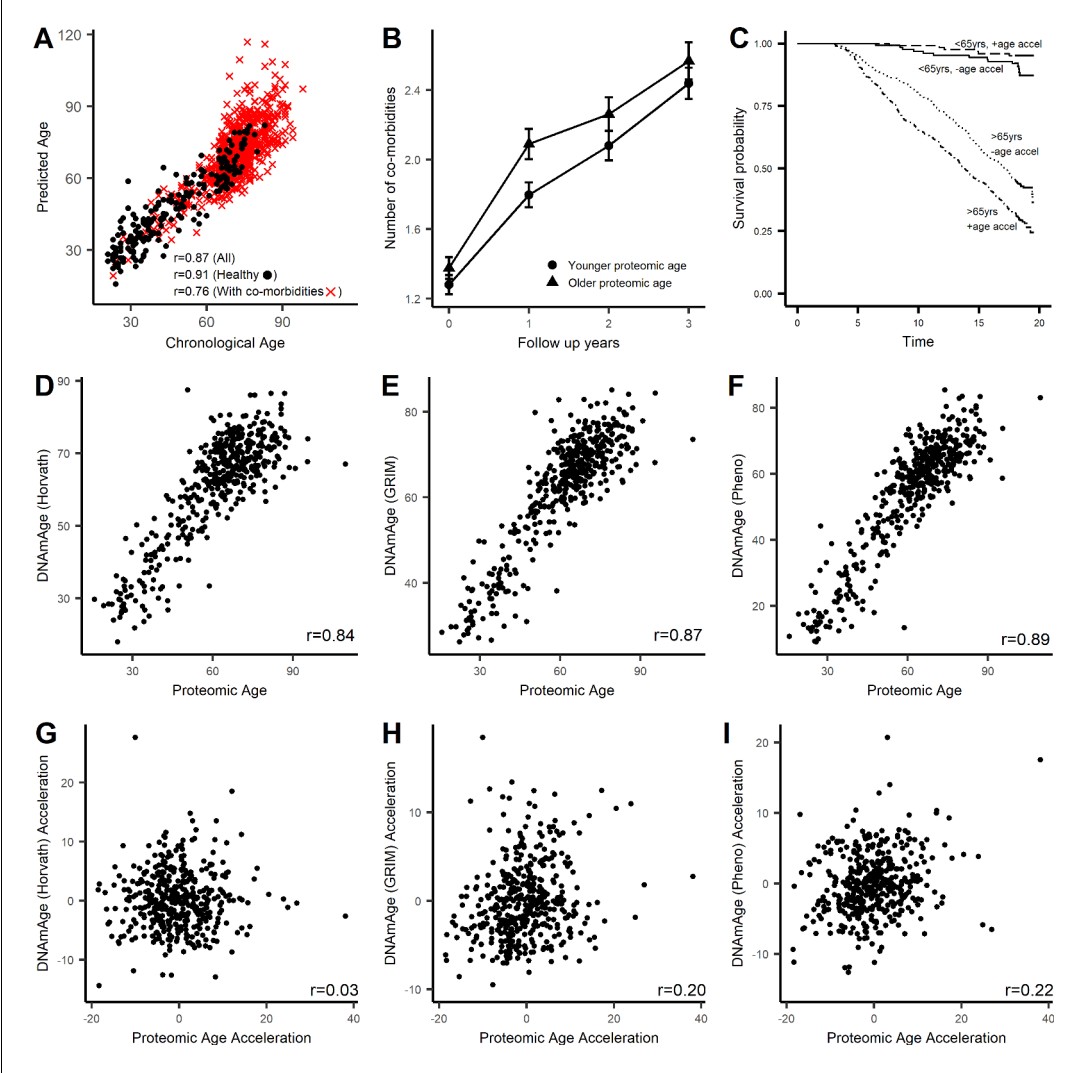

**Figure 8.** Characterization of proteomic signature of age. A proteomic age predictor was created using 76 proteins and a measure of protein biological age was calculated as the residuals (PROaccel) from the regression model of chronological age with predicted age (PROage; *Supplementary file 1D*). (A) Scatter plot of PROage with chronological age. The correlation was higher in subjects free of disease (circle) compared to subject with co-morbidities (cross). Subjects that were greater PROaccel display greater increase in multimorbidity (B) and in higher risk for all-cause mortality in subjects older than 65 years of age (C). The proteomic age was compared with existing epigenetic age clocks. The proteomic signature of age was consistent with other aging clocks including methylation age (*Horvath, 2013*) (D), GRIMage (*Lu et al., 2019*) (E), and phenotypic age (*Levine et al., 2015*) (F). Age acceleration or measure of biological age based on proteomic signature was not correlated with biological aging measured by methylation clock (G), but was correlated with biological age based on GRIMage (H) and phenotypic age (I).

The online version of this article includes the following figure supplement(s) for figure 8:

**Figure supplement 1.** Comparison of proteomic signature of age.

**Figure supplement 2.** Aging rates with trajectory of chronic disease accumulation in healthy disease-free subjects.

To determine whether *PROage* can be used as a measure of 'biological' age, we examined its association with two important age-related phenotypes: number of coexisting chronic diseases (multimorbidity) and all-cause mortality. Both *PROage* and chronological age were significantly associated with the number of coexisting diseases (p<0.05). We considered the measure of biological age as the residuals (*PROaccel*) from the regression line between *PROage* and age. Having a positive *PROaccel* would indicate that the individual is older than their chronological age based on proteomic data. We categorized the subjects as fast- vs slow-agers based on having positive or negative *PROaccel* values, respectively. In subjects over 65 years, there was an average increase of 0.21

comorbidities per year. And in subjects who were 'biologically' older there was greater increase in number of comorbidities by 0.03 diseases per year compared to subjects who were biologically younger (p=1.17×10$^{-5}$; *Figure 8B*). The association of *PROaccel* remained significant and the point estimate remained at an annual change of 0.03 after account for non-linear effects with the inclusion of quadratic and/or cubic term for age in the model. We further tested is *PROaccel* was predictive of developing chronic diseases in subjects free of disease at baseline. We found that independently of chronological age, being biologically older was associated with faster accumulation of age-related diseases by 0.05 diseases per year (p=0.004; *Figure 8—figure supplement 2*).

Both chronological age (HR = 1.14, 95% CI 1.13–1.16) and *PROage* (HR = 1.08, 95% CI 1.07–1.09) were individually associated with all-cause mortality. Independently of chronological age, *PROaccel* was associated with all-cause mortality (HR = 1.03, 95% CI 1.02–1.04). As with multimorbidity, the effect of *PROaccel* was only observed in older subjects. Experiencing accelerated aging was associated with higher risk of mortality in those who were over 65 years (HR = 1.03, 95% CI: 1.02–1.04, p=9.18×10$^{-8}$) but no differences were observed in those under 65 years (HR = 1.00, 95% CI: 0.93–1.09, p=0.905; *Figure 8C*).

To date, several methylation-based measures of biological age have been developed. In a subset of individuals (N = 459), we had data on both *PROage* and three methylation age variables: methylation age (Horvath) (*Horvath, 2013*), GrimAge (*Lu et al., 2019*), and phenotypic age developed by *Levine et al., 2018*. There was high correlation between *PROage* and mDNA$_{HORVATH}$ (r = 0.84, p<0.001, *Figure 8D*), mDNA$_{GRIM}$ (r = 0.87, p<0.001, *Figure 8E*), and mDNA$_{PHENO}$ (r = 0.89, p<0.001, *Figure 8F*). Age acceleration based on proteomic data (*PROaccel*) was not correlated with age acceleration based on the Horvath clock (mDNA$_{HORVATHACCEL}$, r = 0.03, p=0.487, *Figure 8G*), and mildly correlated with mDNA$_{GRIMACCEL}$ (r = 0.20, p<0.001, *Figure 8H*) and mDNA$_{PHENOACCEL}$ (r = 0.22, p<0.001, *Figure 8I*). After adjusting for chronological age, sex and study site, *PROaccel* (HR$_{SD}$ = 1.29; 1.11–1.50, p=0.001,cstatistics = 0.781), mDNA$_{PHENOACCEL}$ (HR$_{SD}$ = 1.32; 1.14–1.54, p=0.0002, cstatistics = 0.782), and mDNA$_{GRIMACCEL}$ (HR$_{SD}$ = 1.44; 1.20–1.74, p=0.0001, cstatistics = 0.781) were associated with all-cause mortality, but mDNA$_{HORVATHACCEL}$ (HR$_{SD}$ = 1.02; 0.88–1.19, p=0.819) was not. In a multivariate model including the four age acceleration metrics, *PROaccel* (1.17; 1.00–1.38), mDNA$_{PHENOACCEL}$ (1.20; 1.02–1.42), and mDNA$_{GRIMACCEL}$ (1.31; 1.07–1.60) were significantly predictive of all-cause mortality, but mDNA$_{HORVATHACCEL}$ (0.94; 0.80–1.11) was not.

## Discussion

A proteomic analysis of chronological age was conducted to identify individuals who are experiencing accelerated health deterioration with aging based on their proteomic profile. The proteomic analysis identified 651 age-associated plasma proteins, many of which had a strong age-relationship independent of the presence of chronic diseases. Many of these proteins were previously described, 224 additional novel age-associated proteins are reported in the InCHIANTI study. Sex-stratified analysis showed that for most proteins, age associations were similar in men and women. Proteins that systematically change with aging are also biomarkers of health longitudinally and risk biomarkers for mortality and accumulation of multimorbidity over time in all subjects as well in persons that appear to be healthy by being free of any chronic diseases at the time of the evaluation. Analysis of DNA methylation data showed that methylation is one of the molecular mechanisms of age-associated differences in protein expression. Exploration of publicly available genetic data to assess the genetic relationship between age-associated protein and aging disease supported the clinical importance of these proteins. Finally, age acceleration measure using proteomic signature of age, which can be interpreted as a proxy measure for biological age, is associated with comorbidity and predicts future rate of change in multimorbidity as well as all-cause mortality over a 18 year follow-up period. These data support the notion that circulating proteins are useful biomarkers to monitor trajectories of health and to identify those individuals who are aging faster than their chronological age. These individuals may be targeted for in depth diagnostic workup to identify novel opportunities for prevention of the impending deterioration of health. Not only are plasma proteins ideal clinical aging biomarker candidates as they are easily accessible, proteins are direct biological effectors and can highlight important mechanisms underlying aging.

Results from several studies that have investigated the relationship between plasma protein and age are consistent in their findings (*Tanaka et al., 2018*; *Sun et al., 2018*; *Menni et al., 2015*; *Lehallier et al., 2019*). The current study in a general Italian population yielded results highly correlated with those from our previous study in a highly selective healthy population from the US, and therefore highlighting that this cluster of age-related proteins is not population specific but rather reflects some general physiological mechanisms of aging, which may have important clinical implications. In our study, 32.6% of the age-associated proteins were predictive of all-cause mortality and 45.3% were associated with multimorbidity independent of chronological age, suggesting that their change may reflect a shared biological process between aging and disease. This observation is consistent with the geroscience paradigm that accelerated aging has a causative role in multiple chronic diseases (*Sierra and Kohanski, 2017*). The protein most strongly associated with chronological age in the current study was pleiotrophin (PTN), which is a heparin-binding growth factor involved in angiogenesis, tumorigenesis, and neuromodulation (*Bertram et al., 2019*; *Souttou et al., 1998*). In our study, PTN was not predictive of either mortality or multimorbidity after adjusting for age. The second most significant protein in our analysis was growth-differentiation factor 15 (GDF15), a member of the transforming growth factor-β cytokine superfamily. GDF15 was the most significant age-associated protein predictive of mortality and multimorbidity. In humans, GDF15 has been suggested as a biomarker of cardiovascular disease mortality, but more recently associated with all-cause mortality (*Lindholm et al., 2018*; *ActiFE study group et al., 2019*). There is growing evidence in both human and animal models that GDF15 plays an important role beyond cardiovascular disease and has been linked to weight homeostasis, fibrosis, diabetes, cardiovascular disease, renal disease, mobility disability and cancer (*Baek and Eling, 2019*; *Bidadkosh et al., 2017*; *Daniels et al., 2011*; *Ho et al., 2013*; *Wallentin et al., 2013*; *Osawa et al., 2020*).

More recently, GDF15 has been identified as a highly robust senescence-associated secretory phenotype (SASP) factor in multiple senescence inducers and cell types (*Basisty et al., 2020*) as well as a driver of senescence-associated colon cancer metastasis (*Guo et al., 2019*). Increase of cellular senescence burden is hypothesized to be one of the fundamental biological mechanisms underlying aging, thus confirmation of GDF15 as a SASP factor (*Basisty et al., 2020*) supports the utility of this protein as a global biomarker of health and lifespan. We have previously reported an enrichment of SASP factors among circulating protein biomarkers of aging (*Tanaka et al., 2018*). Here we have also identified numerous SASP components or other senescence-associated proteins among biomarkers of aging, mortality, and multimorbidity. Among the most robust SASP components we identified 11 proteins in this study - GDF15, MMP1, STC1, IGFBP2, IGFBP4, TIMP1, TIMP2, CST3, CTSB, CLEC11A, and POSTN – which are both core SASP components (secreted by multiple types of senescent cells and senescence inducers) and also associated with all-cause mortality and multimorbidity. IL-6, a classical SASP factor in multiple senescent cell types (*Coppé et al., 2008*), was also associated with all-cause mortality. Several other top SASP components that we identified in plasma are also previously reported health biomarkers (*Basisty et al., 2020*). In addition to aging, MMP1 is a biomarker for several cancers, pulmonary fibrosis and potentially Alzheimer's disease (*Bhat et al., 2012*; *Chen et al., 2016b*; *Rosas et al., 2008*), whereas STC1 is a diagnostic and prognostic biomarker for cancers, pulmonary fibrosis, renal ischemia/reperfusion injury and Alzheimer's disease (*Chang et al., 2015*; *Ohkouchi et al., 2015*; *Pan et al., 2015*; *Shahim et al., 2017*).

The recent INTERVAL and LonGenity studies show that there were unique non-linear patterns in the expression proteins across lifespan (*Lehallier et al., 2019*). More specifically there were three waves of changes in the proteome in the fourth, sixth and eight decade. Moreover, it was shown that the proteins that changed in the older decades were associated with age-related diseases while the proteins that change in the fourth were not. We explored whether similar patterns can be observed in the InCHIANTI study, and strikingly we found a peak shift in the proteome in the eighth decade at 78 years of age replicating the same exact timing of the INTERVAL and LonGenity study. T in the younger ages, however our study may have been limited in identifying a wave in ages before 60 due to the small sample size resulting from the sampling design in the InCHIANTI study. We also found that the proteins that were associated with age in the older years were associated with both all-cause mortality and multimorbidity. In our study, the proteins identified through linear modeling were also enriched in proteins associated with all-cause mortality and multimorbidity thus the linear models did capture important aging biomarkers. However, these data suggest that more emphasis

on proteins that change in later years may lead to discovery of clinically important proteomic biomarkers of age.

MR explores causal relationships between exposure and outcome using the random assortment of genetic alleles associated with exposures of interest, often described as a 'natural' RCT study. Two sample MR utilizes large databases of genome-wide association study results to infer causal relationships using genetic data as instrument variables. In this study, we find that genetically elevated brain type natriuretic peptide (NPPB) was protective against hypertension. Circulating NPPB reduces blood pressure and plasma volume through its effect on the adrenal gland, kidney, brain and vasculature (*Levin et al., 1998*). Circulating NPPB levels are found at high concentrations in patients with heart failure and was shown to be a useful diagnostic tool in an acute setting (*Davis et al., 1994*; *Gardner, 2003*). A genome-wide association study of circulating NPPB identified a variants in the NPPB locus, and these variants were also associated with lower systolic and diastolic blood pressure as well as lower risk of hypertension (*Newton-Cheh et al., 2009*). The genetic instrument for NPPB used in our study is mapped to the same locus as the variants used in the previous report, and our results are consistent with their findings. We also found genetically higher matrix metalloproteinase-12 (MMP12) to be protective of coronary heart disease, MI, and ischemic stroke. MMP12 is a metalloprotease produced primarily by macrophages and is involved in inflammation and tissue damage (*Shapiro et al., 1993*). While circulating levels of MMP12 have been linked to increased risk of cardiovascular disease, genetically increased MMP12 has been linked to reduced cardiovascular disease risk in the current report as well as others (*Sun et al., 2018*; *Chong et al., 2019*). The apparent discrepant association between the genetic instrument for NPPB and MMP12 with their respective age-associated disease may signify reverse-causation where proteins levels are altered in response to development of disease or may reflect possible compensatory protein where protein levels increase with age to protect against development of disease (*Bucur et al., 2020*). The discrepancy of the direction of associations should be explored further. We found genetically increased proteinase inhibitor 3 (PI), or elafin, was associated with increased odds of having MI, both ischemic and cardioembolic stroke, lunch cancer and rheumatoid arthritis. PI3 is a skin-derived serine protease inhibitor that plays an important role in hydrolyzing connective tissue components including elastin and collagen through the inhibition of elastase (*Schalkwijk et al., 1990*; *Wiedow et al., 1990*). PI3 has also been linked to lung tissue repair and is a candidate biomarker for lung diseases including acute respiratory lung disease and asthma (*Wang et al., 2017*; *Tsai et al., 2016*), as well as modulator of inflammation through the inhibition of neutrophil derived elastases (*Alam et al., 2012*). Our data support the involvement of PI3 in cardiovascular, lung and immunological diseases through the control of inflammation and tissue repair. We found genetically higher SPARC related modular calcium binding 1 (SMOC1) to be associated with higher bone mineral density, hemoglobin and RDW. Our results are consistent with results from genome-wide association studies of bone mineral density and osteoporosis, where SMOC1 has been identified as the candidate gene underlying the signal on chromosome 14 (*Zhang et al., 2014*; *Qiu et al., 2019*). Animal studies have shown that SMOC1 knockout results in disruption of ocular and limb development through reduction in interdigital apoptosis and disruption in BMP signaling. Interestingly, SMOC1 was identified in a genome-wide association study of blood traits in mice, however, this locus has not been a major locus in human genetic studies of blood parameters (*Davis et al., 2013*). Further studies are needed to explore the link between SMOC1 and blood traits found in our study. Finally, we find genetically higher growth hormone receptor (GHR) was associated with bone mineral density. GHR knockout animals display disproportional bone growth and lower bone mineral density (*Sjögren et al., 2000*). In humans, administration of growth hormone has been shown to improve bone mineral density in postmenopausal women (*Joseph et al., 2008*).

Our data suggest that the proteomic signature of age (*PROage*) can be effectively used to assess an individual's rate of aging (*PROaccel*). What was surprising is that the proteomic age signature developed by our group is strikingly similar to another proteomic age signature created in a geographically distinct population by *Lehallier* and colleagues (*Lehallier et al., 2019*). This suggests that the circulating age proteome, at least as captured by this aptamer-based platform, is very robust. Using this proteomic signature, subjects who were predicted to be older than their chronological age had higher prevalence of comorbidities, more rapid increase in co-morbidities over time, and higher risk of all-cause mortality. In recent years, several measures of rates of aging have been developed using methylation data. We compared the proteomic signature with three epigenetic clocks.

One of the first phase of epigenetic clocks developed by *Horvath, 2013*, is a multi-tissue methylation signature (mDNA$_{HORVATH}$) that was designed to predict chronological age. The second is an epigenetic marker for phenotypic age developed by Levine and colleagues (*Levine et al., 2018*) that was derived using clinical biomarkers, such as albumin, creatinine, and glucose (mDNA$_{PHENO}$). Phenotype age was designed to predict all-cause mortality, cause-specific mortality, and other aging measures. The final epigenetic clock is one developed by Ake and colleagues (*Lu et al., 2019*) based on biomarkers and risk factors of mortality including smoking, seven biomarkers including GDF15 and PAI-I (mDNA$_{GRIM}$). Interestingly, aging rate based on proteomic clock (*PROaccel*) was not correlated with aging rates based on mDNA$_{HORVATH}$ (mDNA$_{HORVATHACCEL}$), but was moderately correlated with rates based on mDNA$_{PHENO}$ (mDNA$_{PHENOACCEL}$) and mDNA$_{GRIM}$ (mDNA$_{GRIMACCEL}$). Since both *PROaccel* and mDNA$_{GRIMACCEL}$ both include GDF15, these two biological aging measures may be capturing similar aging mechanisms. We compared the *PROaccel* against three methylation age measures in their ability to predict all-cause mortality. *PROaccel* performed better than mDNA$_{HORVATHACCEL}$, but was outperformed by mDNA$_{PHENOACCEL}$ and mDNA$_{GRIMACCEL}$. This would suggest that creating a proteomic signature of aging phenotypes, rather than chronological age, could result in a more robust proteomic biomarker for aging and should be explored in future studies. However, our biomarker panels include many SASP proteins that may be utilized as important benchmarks for clinically assessing senescent cell burden. As the development of drugs that selectively eliminate senescent cells (senolytics) or modify the SASP (senomorphics) has quickly advanced in recent years, with early trials now in progress, using biomarker panels that include markers of senescence will be an important step in translating these interventions to the clinic.

There are several limitations to this study. While the associations of individual proteins with age have been consistent across different populations, the associations of the *PROaccel* with morbidity and mortality should be confirmed in additional, larger studies. It would also be important to validate whether *PROage* measure can be confirmed using proteomic data generated using different platforms. Since proteomic measurements were only conducted cross-sectionally, we could not determine whether *PROaccel* is a sensitive measure of changes in health status over time. Longitudinal measurement of proteomic data will also be useful determining the trajectories of protein abundances, which could be a more sensitive in capturing rates of aging.

In summary, in this study 651 age-associated proteins were identified that are also predictive of important aging outcomes including all-cause mortality and multimorbidity. Further examination of the function of these proteins individually or collectively may further shed light to the mechanisms underlying the aging process. Finally, a proteomic signature of age based on abundances of 76 proteins captures the rate of aging and may be useful in identifying individuals who are aging faster than their chronological age. These findings should be confirmed in a longitudinal perspective, with good representation of individual men and women across the lifespan. Finally, clinical studies that use in depth characterization of physiological and pathologic characteristics of individuals who are 'older' compared to those who are 'younger' than their chronological aging may confirm whether a proteomic signature of aging may be a useful tool in clinical practice.

## Materials and methods

### Subjects and methods

The InCHIANTI study is a community-based cohort study based in the Tuscan region of Italy. This study was conducted in the 'Invecchiare in Chianti' (Aging in the Chianti Area, InCHIANTI) study. The InCHIANTI study is a population-based epidemiological study aimed at evaluating factors that influence mobility in the older population living in the Chianti region in Tuscany, Italy. Details of the study have been previously reported (*Ferrucci et al., 2000*). Briefly, 1616 residents were selected from the population registry of Greve in Chianti (a rural area: 11,709 residents with 19.3% of the population older than 65 years of age) and Bagno a Ripoli (Antella village near Florence; 4704 inhabitants, with 20.3% older than 65 years of age). The participation rate was 90% (n = 1453) and subjects' age ranged between 21 and 102 years. Overnight fasted blood samples were used for genomic DNA extraction and genotyping. The study protocol (exemption #11976) was approved by the Italian National Institute of Research and Care of Aging Institutional Review and Medstar

Research Institute (Baltimore, MD) and approved by the Internal Review Board of the National Institute for Environmental Health Sciences (NIEHS). All participants provided written informed consent.

Sociodemographic information (age, sex) was obtained during a structured interview. Mortality data were evaluated for up to 20 years following the baseline visit (1998–2000), assessed using the mortality general registry maintained by the Tuscany Region and through death certificates submitted to the Registry office of the municipality of residence immediately after death (*Zuliani et al., 2017*). Whole blood white blood cell differential count was assessed using a Coulter Counter (LH 750 Hematology Autoanalyzer, Beckman Coulter Inc, Brea, CA). Multimorbidity was ascertained based on the presence of 15 diseases (hypertension, diabetes, ischemic heart disease, congestive heart failure, stroke, chronic obstructive pulmonary disease, cancer, Parkinson's disease, hip fracture, lower extremities joint disease, anemia, PAD, cognitive impairment, chronic kidney disease, and depression) as previously described (*Fabbri et al., 2015*).

## Proteomic assessment

Proteomic profiles for 1322 SOMAmers were assessed using the 1.3 k SOMAscan Assay at the Trans-NIH Center for Human Immunology, Autoimmunity, and Inflammation (CHI), National Institute of Allergy and Infectious Diseases, National Institutes of Health (Bethesda, MD, USA) using the same methods as previously published (*Tanaka et al., 2018*). Out of the 1322 SOMAmer reagents, 12 hybridization controls, four viral proteins (HPV type 16, HPV type 18, isolate BEN, isolate LW123) and 5 SOMAmers that were reported to be non-specific (P05186; ALPL, P09871; C1S, Q14126; DSG2, Q93038; TNFRSF25, Q9NQC3; RTN4) were removed, thus leaving 1301 SOMAmer reagents in the final analysis. There are 46 SOMAmer reagents that target multicomplex proteins of two or more unique proteins. There are 49 UniProt IDs that are measured by more than one SOMAmer reagent. Thus, the 1301 SOMAmer reagents target 1297 UniProt IDs. Of note, there are four proteins (P05413; FABP3, P48788; TINNI2, P19429; TINNI3, P01160; NPPA) in the final protein panel that are rat homologues of human proteins.

The experimental process for proteomic assessment and data normalization has been previously described (*Candia et al., 2017*; *Cheung et al., 2017*). The data reported are SOMAmer reagent abundance in relative fluorescence units (RFU). The abundance of the SOMAmer reagent represents a surrogate of protein concentration in the plasma sample. Data normalization was conducted in three stages. First, hybridization control normalization removes individual sample variance on the basis of signaling differences between microarray or Agilent scanner. Second, median signal normalization removes inter-sample differences within a plate due to technical differences such as pipetting variation. Finally, calibration normalization removes variance across assay runs. Furthermore, there is an additional inter-plate normalization process that utilizes CHI calibrators that allows normalization across all experiments conducted at the CHI laboratory (*Candia et al., 2017*; *Cheung et al., 2017*).

## Genome-wide DNA methylation assessment

Details of DNA methylation assessment and QC filtering in the InCHIANTI study have been described elsewhere (*Moore et al., 2016*). In brief, genomic DNA was extracted from buffy coat samples (white blood cells) collected at the baseline visit and methylation status of ~480,000 CpG were assessed with Illumina Infinium HumanMethylation450 BeadChip (Illumina Inc, San Diego, CA) using manufacturer's protocol. Quality filtering (bead count, detection rate) and normalization was performed using the DASEN method in the watermelon R package (*Pidsley et al., 2013*). After filtering CpG with cross-reactive probes, on polymorphic position, and on X or Y chromosomes, 472,139 markers in 460 subjects with both proteomic and methylation data were used for analysis.

## Statistical analysis

Unless noted otherwise, statistical analysis was performed using R, a language and environment for statistical computing and graphics. Protein RFU abundances were natural-log transformed and outliers outside 4SD were removed. Associations of protein with chronological age were assessed by linear regression using the lm function adjusting for sex and study site (base model). A second model was examined with further adjustments for the presence of chronic diseases. The proportion of variation explained by age was calculated as the difference between the adjusted R-square between a model with and without age, while adjusting for sex and study site. To test for differences in age-

protein association by sex, an age-by-sex interaction term was included in the base model. For all analyses, a Benjamini–Hochberg false discovery rate (FDR) was applied using the p.adjust function to adjust for multiple testing. Associations were considered significant at FDR < 0.05. A meta-analysis of results from the BLSA/GESTALT study (*Tanaka et al., 2018*) and InCHIANTI study was performed using a fixed effect inverse-variance meta-analysis using the command-line tool METAL (*Willer et al., 2010*). Differential expression or non-linear association of proteins with age was conducted using Differential Expression - Sliding-Window ANalysis (DE-SWAN) R package (*Lehallier et al., 2019*) using an age window of 10, 20 and 30 years. A q-value threshold of $\leq 0.05$ was considered significant.

Association of protein with all-cause mortality was analyzed using follow-up time to death as the time variable. Cox proportional hazard models were assessed adjusting for covariates (age, sex, and study site) using the coxph function from the survival package. For each fully adjusted model, the proportional hazards assumption was tested by examining the correlation of scaled Schoenfeld residuals against time using the cox.zph function. A global p-value$\leq 0.05$ was considered a violation of proportional hazard assumption (26 proteins). For these proteins, a stratified analysis was conducted by splitting the follow-up period into two or three intervals and including an interaction term between protein and time-interval strata in the model. Based on the plots of the Schoenfeld residuals versus transformed time, follow-up time was split into two intervals at the mean follow-up year (15 years) for 18 proteins, or into 3 intervals at 12 and 18 years ($1^{st}$ and $3^{rd}$ quarter) for the remaining eight proteins. To investigate the association of age-related protein with trajectories of multimorbidity, the slope was extracted from a linear mixed model that repeated measurement of number of multi-morbidities over a 10 year period. The associations between slope and age-related protein levels were modeled using linear regression adjusting for covariates. In addition to the single-protein model, for both all-cause mortality and multimorbidity, we used the least absolute shrinkage and selection (LASSO) method (implemented via the glmnet function) to perform variable selection and construct parsimonious multi-protein models. To compare the effect sizes of each protein on mortality or multimorbidity, protein abundances were z-transformed to zero mean and unit variance.

## Mendelian randomization

To explore the causal relationship between age-related plasma proteins with chronic diseases, we used a two-sample MR approach. This is accomplished by identifying genetic instrument variable for the exposure (plasma protein levels) and outcome variable (chronic diseases) the estimating the causal effect. The plasma proteins that were associated with all-cause mortality and multimorbidity in the multi-protein models (protein list here) were targeted for MR analysis. Protein quantitative trait loci (pQTL) for these plasma protein levels were selected from a pQTL analyses conducted in the INTERVAL study (*Sun et al., 2018*). For all proteins except GDF15, there was only one genetic variant that was significantly associated with protein levels; NPPB (rs198389), GDF15 (rs45543339, rs1227734), MMP12 (rs28381684), PI3 (rs16989763), GHR (rs150036324), TNFRSF1B(rs5746017), and SMOC1(rs1958078; Table S4). The chronic disease outcomes tested included 18 trait/diseases (hypertension, type 2 diabetes, ischemic heart disease, coronary heart disease, ischemic heart disease, cardioembolic stroke, chronic kidney disease, prostate cancer, lunch cancer, ovarian cancer, Parkinson's disease, depressive disorder, Alzheimer's disease, rheumatoid arthritis, spine bone mineral density, hemoglobin, bone mineral density, red blood cell distribution width) that reflect the 15 diseases used to ascertain multimorbidity status (*Hemani et al., 2018*). For genetic instrument with one genetic variant, the causal effect was estimated using the Wald ratio (*Lawlor et al., 2008*). For GDF15, where two genetic variants comprised the genetic instrument, an inverse-variance weighted (IVW) linear regression was used (*Bowden et al., 2015*). All analysis was conducted using the two-sampleMR package in MR-base (*Hemani et al., 2018*).

## Mediation analysis using methylation

Baron and Kenny method was used to test whether age-associated changes in protein levels are mediated by DNA methylation (*Baron and Kenny, 1986*) in a subset of 460 individuals with both proteomic and genome-wide methylation data. The residuals of proteins and methylation values after adjustment for key covariates were used for this analysis. That is, for protein RFU, residuals after adjustment for sex and study site, and for methylation residual after adjustment for sex, study

site, batch, and percentage of neutrophils, lymphocytes, monocytes, eosinophils, and basophils were used. For all age-associated proteins, DNA methylation site (CpGs) positioned 10 kb within each gene (cis-CpG) were tested for association with age. Enrichment of age-associated CpG was tested using fisher's exact test. The age-associated CpG identified were used in subsequent mediation analyses. In the first step, age-associated proteins were identified by regressing protein residuals on age using a linear regression model. In the second step, age-associated CpG methylation was identified by regressing methylation residuals on age using a linear model. In the final step, a linear model regressing protein residual on age adjusting for methylation residuals was tested. Significance of the mediation effect was tested using the Sobel test (*Sobel, 1982*). A non-significant direct effect of age on protein in the final model (adjusted for methylation) was considered as complete mediation, while a significant direct effect in the final model was considered as partial mediation.

## Proteomic signature of age

A predicted proteomic age (*PROage*) signature was constructed as a linear combination of the regression coefficients for 76 proteins from the penalized regression model that was previously published (*Tanaka et al., 2018*). A measure of proteomic biological age was calculated as the residuals from the regression model of chronological age with PROage (*PROaccel*). Individuals with positive *PROaccel* are considered to have accelerated aging, while those with negative *PROaccel* are considered to be aging slower. The association of *PROaccel* with the number of multimorbidity at baseline was assessed using Poisson regression. The association of *PROaccel* with all-cause mortality was assessed using Cox regression.

## Pathway enrichment and visualization

Gene set over-representation analyses were performed using the ConsensusPathDB-human tool, release 34 (15.01.2019) (*Kamburov et al., 2013*). Curated pathways for enrichment analysis were referenced from the following databases: Gene Ontology, WikiPathways, and Kegg pathways, and Reactome. Pathways were required to have a minimum of three observed proteins and a p-value<0.01. Gene ontology terms were restricted to levels 3–5. For morbidity-associated proteins (*Figure 5A*), all pathways with enrichment q-values <0.05 were kept, and the top 25 pathways were plotted. For mortality-associated proteins (*Figure 4A*), all pathways with enrichment q-values <0.25 were kept. One duplicate named pathway appearing from multiple source databases was omitted manually. For background reference, a list of all 1301 proteins measured in the SOMAscan assay were used. Tables containing the pathways, observed corresponding proteins, reference pathway annotations, statistics, and source databases are included as a supplementary file (*Supplementary file 1E*).

Heatmaps and dot plots were generated in R with the 'ggplot2' package (*Wickham et al., 2016*) and by comparing with SASP profiles on SASP Atlas (*Basisty et al., 2020*) (http://www.saspatlas.com/). Color palettes in R were generated with the 'RColorBrewer' package (*Neuwirth, 2014*).

## Acknowledgements

The InCHIANTI study baseline (1998–2000) was supported as a 'targeted project' (ICS110.1/RF97.71) by the Italian Ministry of Health and in part by the US National Institute on Aging (Contracts: 263 MD 9164 and 263 MD 821336). This work was supported by grants from the National Institute on Aging (U01 AG060906, PI: Schilling and K99 AG065484, PI: Basisty). This work utilized the computational resources of the NIH HPC Biowulf cluster. (http://hpc.nih.gov).

## Additional information

### Funding

| Funder | Grant reference number | Author |
|---|---|---|
| National Institutes of Health | U01 AG060906 | Birgit Schilling |
| National Institutes of Health | K99 AG065484 | Nathan Basisty |

The funders had no role in study design, data collection and interpretation, or the decision to submit the work for publication.

## Author contributions

Toshiko Tanaka, Conceptualization, Formal analysis, Visualization, Writing - original draft, Writing - review and editing; Nathan Basisty, Formal analysis, Visualization, Writing - review and editing; Giovanna Fantoni, Data curation, Writing - review and editing; Julián Candia, Data curation, Formal analysis; Ann Z Moore, Birgit Schilling, Writing - review and editing; Angelique Biancotto, Data curation; Stefania Bandinelli, Data curation, Supervision, Writing - original draft; Luigi Ferrucci, Conceptualization, Supervision

## Author ORCIDs

Toshiko Tanaka  https://orcid.org/0000-0002-4161-3829
Julián Candia  https://orcid.org/0000-0001-5793-8989
Birgit Schilling  http://orcid.org/0000-0001-9907-2749
Luigi Ferrucci  http://orcid.org/0000-0002-6273-1613

## Ethics

Human subjects: The study protocol (exemption #11976) was approved by the Italian National Institute of Research and Care of Aging Institutional Review and Medstar Research Institute (Baltimore, MD) and approved by the Internal Review Board of the National Institute for Environmental Health Sciences (NIEHS).

## Decision letter and Author response

Decision letter https://doi.org/10.7554/eLife.61073.sa1
Author response https://doi.org/10.7554/eLife.61073.sa2

# Additional files

## Supplementary files

• Source code 1. R code for primary analyses in this manuscript.

• Supplementary file 1. Supplemental tables. (1A) Age-associated proteins mediation by DNA methylation (1B) Age-associated proteins at different age waves (1C) Age-associated diseases considered for Mendelian randomization (1D) 76-protein signature weights (1E) Enrichment analysis of proteins associated with mortality and multimorbidity

• Transparent reporting form

## Data availability

Phenotypic data and source codes used for this manuscript are provided. Due to the contents of the InCHIANTI study consent forms, proteomic and DNA methylation data cannot be made publicly available. Researchers can seek access to these data through the submission of proposals and subsequent approval through the InCHIANTI study website (http://inchiantistudy.net/).

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
