## [Decision Letter]

**Acceptance summary:**

The work highlights the importance of proteomic biomarkers in aging and its association across other data types.

**Decision letter after peer review:**

Thank you for submitting your article "Plasma proteomic biomarker signature of age predicts health and life span" for consideration by *eLife*. Your article has been reviewed by three peer reviewers, and the evaluation has been overseen by a Reviewing Editor and Anna Akhmanova as the Senior Editor. The reviewers have opted to remain anonymous.

The reviewers have discussed the reviews with one another and the Reviewing Editor has drafted this decision to help you prepare a revised submission.

As the editors have judged that your manuscript is of interest, but as described below that additional analyses are required before it is published, we would like to draw your attention to changes in our revision policy that we have made in response to COVID-19 (https://elifesciences.org/articles/57162). First, because many researchers have temporarily lost access to the labs, we will give authors as much time as they need to submit revised manuscripts. We are also offering, if you choose, to post the manuscript to bioRxiv (if it is not already there) along with this decision letter and a formal designation that the manuscript is "in revision at *eLife*". Please let us know if you would like to pursue this option. (If your work is more suitable for medRxiv, you will need to post the preprint yourself, as the mechanisms for us to do so are still in development.)

Summary:

This is a well-written article that analyzes a proteomics data set of a large number of individuals and conduct several analyses to identify protein biomarkers of aging, mortality and aging related diseases. They also integrate the result with genetic and other data. The work adds to the growing field of biomarkers of aging and related diseases. There are, however, some concerns about statistical methods used that need attention before the manuscript can be accepted for publication in *eLife*.

Essential revisions:

1) A general concern is that it is not clear what the authors controlled in various hypothesis tests. Different names and possibly different methods were presented for statistical significances as "FDR", "FDR-q", "p", "P", "pfdr", "FDR_P", and so on. Multiple testing correction was not likely applied in some analyses, for examples, mediation analysis and Mendelian randomization. It would be much easier to understand the results if only one method, e.g. B-H FDR, and label was used for all analyses.

2) The authors report an analysis of sex specific aging proteins. It would be important to replicate the results in an independent population. The interval study should have data to replicate the sex-specific aging signature.

3) Mediation by methylation. This analysis is interesting but there are some points that need clarification. Specifically, how did the author test for the enrichment of the age-associated CpG within 10 kb of aging proteins? Also, was the mediation tested only for CpG sites near genes, so only "cis" relations were tested but not "trans" relations? Finally, the claim about ENNP7 would be more believable if the authors show the relation between protein data versus the methylation data for this gene.

4) Association of age-related proteins with multimorbidity. A clean analysis would model the number of diseases using a Poisson regression and there is no need to run a two steps analysis, the authors could model the number of diseases over time using mixed effect models, or more easily GEE.

5) Cox proportional hazards models. The time variable of the models was not clarified. A crucial assumption of the Cox models, proportional hazards, was not checked by a test such as Schoenfeld residuals.

6) Differential protein expression across age. The plots show that the distribution of their population is not uniform over age. Their plots in Figure 6 shows a very suspicious correlation between the peak of the number of significant proteins and the larger concentration of individuals in their samples. The number of expressed proteins in a set is a function of the sample size, and the claim of the authors about the age pattern of significant proteins is not valid. The effects of the non-linear correlation were not taken into account in other analyses. Because of the effects, the adjustment for age was likely not enough for the Cox models, assuming follow-up time was the time variable. In Figure 8A, PROaccel cannot be independent from chronological age. Possibly this explains why the authors observed a significant mortality association in the older group only. At least quadratic effects of age, possibly higher orders, should be considered.

7) Proteomic signature of age. First, the authors have a proteomic score of age, not a signature, which is essentially a set with patterns. In addition, the analysis of negative and positive residuals is not valid statistically. The authors should use an interval about the predicted chronological age, and then define the groups of slower and faster age based on their statistically significant difference from the predicted age.

8) Pathway enrichment analysis. Some pathways were presented to be overrepresented in the set without any detailed information of the analysis. Background set and annotated sets for each pathway should be described possibly in a table.

9) It is not apparent which part of the findings or observations were new. It seems some parts e.g. aging signatures with 76 proteins, were supportive replications of previous findings of the authors or others. Any discussion about novel findings would be helpful to understand the significance of this paper.

[Editors' note: further revisions were suggested prior to acceptance, as described below.]

Thank you for resubmitting your article "Plasma proteomic biomarker signature of age predicts health and life span" for consideration by *eLife*. Your article has been reviewed and discussed among the reviewers and the Reviewing Editor and Anna Akhmanova as the Senior Editor.

We believe the manuscript has improved but some concerns were still raised that are listed below. These needs to be taken into consideration before the paper can be accepted for publication in *eLife*.

Revisions:

1) It seems the authors simply replaced those various labels with "B-H FDR". Hence, some values in the columns "B-H FDR" cannot be correct. For example, the values of "B-H FDR" are identical to "p" in Supplementary file 1B.

2) No violation of the proportional hazards assumption should be clearly stated in the manuscript. As only the global p-value was given in the response, it is not clear whether the authors had checked the significance of every variable in their Cox model or not.

3) The text "did not affect the results" is too vague. It should be clearly stated with more explanation about the model and results.

4) All genes should not be used as the background reference because those included the genes for the proteins the authors didn't measure at all. When all genes were used as the reference set, such enrichment analysis likely identifies the enrichment among the 1301 measured proteins, which were proteins in plasma, comparing to all proteins. This explains why the authors found strong enrichments of inflammatory pathways.

---

## [Author Response]

Essential revisions:1) A general concern is that it is not clear what the authors controlled in various hypothesis tests. Different names and possibly different methods were presented for statistical significances as "FDR", "FDR-q", "p", "P", "pfdr", "FDR_P", and so on. Multiple testing correction was not likely applied in some analyses, for examples, mediation analysis and Mendelian randomization. It would be much easier to understand the results if only one method, e.g. B-H FDR, and label was used for all analyses.

We have gone through the tables and text to unify the labels as “B-H FDR” where multiple testing was applied.

2) The authors report an analysis of sex specific aging proteins. It would be important to replicate the results in an independent population. The interval study should have data to replicate the sex-specific aging signature.

We have compared our results to those reported by Lehallier *et al.* in the INTERVAL study. Of the proteins that were covered in the INTERVAL study, 78.2% of the sex-associated proteins were replicated. We have included these findings in the results as well as in Figure 2—source data 1.

“To test the robustness of these association, these results were compared to results from INTERVAL study (18). Of the 427 proteins associated with sex in the InCHIANTI study, 294 (68.9%) were measured in the INTERVAL study (Figure 2— source data 1). Of these, 230 proteins (78.2%) were associated with sex in the same direction with the INTERVAL study, reflecting the robustness of sex-associated proteins.”

3) Mediation by methylation. This analysis is interesting but there are some points that need clarification. Specifically, how did the author test for the enrichment of the age-associated CpG within 10 kb of aging proteins? Also, was the mediation tested only for CpG sites near genes, so only "cis" relations were tested but not "trans" relations? Finally, the claim about ENNP7 would be more believable if the authors show the relation between protein data versus the methylation data for this gene.

We appreciate the reviewer pointing out that the analysis was not clear. For the enrichment of age-associated CpG, we ran a fisher’s exact test for CpG within 10kb of the gene. And we only tested for cis, and not trans CpG. These points were clarified in the Materials and methods section. The association of CpG with protein values in the model are shown in Supplementary file 1A (column M-O). The ENNP7 CpG methylation and protein values were negatively associated.

“For all age-associated proteins, DNA methylation site (CpGs) positioned 10kb within each gene (cis-CpG) were tested for association with age. Enrichment of age-associated CpG was tested using fisher’s exact test. The age-associated CpG identified were used in subsequent mediation analyses. “

4) Association of age-related proteins with multimorbidity. A clean analysis would model the number of diseases using a Poisson regression and there is no need to run a two steps analysis, the authors could model the number of diseases over time using mixed effect models, or more easily GEE.

We thank the reviewers for the comment. We initially ran a mixed effects model for the first step for the trajectories of multimorbidity. Therefore, we ran a LASSO variable selection using glmmLasso. However, using this approach we had convergence issues. We therefore took the approach of initially extracting the slope from the mixed model then running a linear model. The results from single protein analysis are comparable between the two-step analysis and single mixed model results therefore we opted to present this model.

5) Cox proportional hazards models. The time variable of the models was not clarified. A crucial assumption of the Cox models, proportional hazards, was not checked by a test such as Schoenfeld residuals.

We agree that the descriptions of the models were lacking. We have clarified the time variable in the Materials and methods section. Before the analysis, we tested the proportionality of the hazards assumption in the base model with covariates age, sex, and site, and found that these assumptions were not violated (global p=0.48).

“In order to explore the association of protein with all-cause mortality, using follow-up time to death as the time variable”

6) Differential protein expression across age. The plots show that the distribution of their population is not uniform over age. Their plots in Figure 6 shows a very suspicious correlation between the peak of the number of significant proteins and the larger concentration of individuals in their samples. The number of expressed proteins in a set is a function of the sample size, and the claim of the authors about the age pattern of significant proteins is not valid. The effects of the non-linear correlation were not taken into account in other analyses. Because of the effects, the adjustment for age was likely not enough for the Cox models, assuming follow-up time was the time variable. In Figure 8A, PROaccel cannot be independent from chronological age. Possibly this explains why the authors observed a significant mortality association in the older group only. At least quadratic effects of age, possibly higher orders, should be considered.

We thank the reviewers for these comments, and we agree that our population by design is weighted with subjects that are older and may not be the ideal population to test for this question. However, we wanted to see if we can observe similar trends that were reported by Lehallier et al. as we found these results to be fascinating. We did observe a similar peak in the 8^th^ decade but not the younger peaks. We do consider issues of power due to sample size at younger ages and include this as a limitation in our analysis in the text.

“There were no clear waves in the younger ages, however our study may have been limited in identifying a wave in ages before 60 due to the small sample size resulting from the sampling design in the InCHIANTI study.”

We included mortality analysis with both quadratic and cubic age effects and the results were unchanged. We included this discussion in the Results section.

“To account for non-linear effects, inclusion of quadratic and/or cubic effects of age did not affect the results.”

7) Proteomic signature of age. First, the authors have a proteomic score of age, not a signature, which is essentially a set with patterns. In addition, the analysis of negative and positive residuals is not valid statistically. The authors should use an interval about the predicted chronological age, and then define the groups of slower and faster age based on their statistically significant difference from the predicted age.

We agree with the reviewers that signature may not be the most accurate description of what we are describing. However, to keep consistency from our previous publication, we would prefer to keep the reference to a “signature” in this new manuscript.

In regards to the use of residuals, we agree that the residuals will not have any biological significance in the training set, as these are random variables with a mean of 0. However, this is only true if all sources of variability in the outcomes measures are considered in the in the regression. Here, we hypothesize that some of the residual variability is related to an unobserved variable that is the “non chronological rate of aging”. Thus, the residuals (or alternatively the difference between the predicted and chronological age) carry some biological meaning. In this manuscript we show that those who are younger in proteomic age compared to chronological age do better than those who are older. We implemented this strategy to complement many papers working on “aging clocks” including the epigenetic clocks. Since the clock is tuned on chronological age, getting an interval around the predicted age and seeing whether the predicted age is significantly lower or higher than the actual age would be difficult since the proteomic clock predicts age with high accuracy, there will probably not be many people who would fall into the slower or faster aging group. This begs the question of whether chronological age is the correct variable for tuning the clock and obtaining information on biological age. This is something that our group is actively discussing, and we are in the process of creating other scores that may capture rates of non-chronological aging more effectively. This is an important point that we discuss in the text.

“This would suggest that creating a proteomic signature of aging phenotypes, rather than chronological age, could result in a more robust proteomic biomarker for aging and should be explored in future studies.”

8) Pathway enrichment analysis. Some pathways were presented to be overrepresented in the set without any detailed information of the analysis. Background set and annotated sets for each pathway should be described possibly in a table.

We thank the reviewers for pointing out this missing information. We have now altered the text in the Materials and methods section to describe the background set used for enrichment analysis and included supplemental tables (Supplementary file 1E) corresponding to the pathway enrichment figures (Figures 4A and 5A) that contain detailed information regarding the pathway names, annotations, statistics, source databases, statistics, total genes, and observed genes.

“Pathway enrichment, network analysis, and visualization

Gene ontology, pathway, and network analysis was performed using the ClueGO package, version 2.5.6, in Cytoscape (https://cytoscape.org/), version 3.7.2 (74, 75). Curated pathways for enrichment analysis were referenced from the following databases: WikiPathways, and Kegg pathways, and Reactome. Pathways were required to have a minimum of three observed proteins. Some redundant or duplicate pathways appearing from multiple source databases were omitted manually. For background reference, the total number of genes associated with all terms included in the ontology source were used. Tables containing the pathways, observed molecules, reference pathway annotations, statistics, and source databases are included as a supplemental table (Supplementary file 1E). The statistical cutoff for enriched pathways was Bonferroni-step-down-adjusted p-values <0.001 by right-sided hypergeometric testing. Pathway-connecting edges were drawn for kappa scores >40%. Kappa scores are a measure of inter-pathway agreement among observed proteins that indicate whether pathway agreement is greater than expected by chance based on shared proteins. Pathways with the same color indicate ≥50% similarity in terms.”

9) It is not apparent which part of the findings or observations were new. It seems some parts e.g. aging signatures with 76 proteins, were supportive replications of previous findings of the authors or others. Any discussion about novel findings would be helpful to understand the significance of this paper.

Thank you for pointing this out. One of the most exciting findings is that measures of proteins (either single proteins or multiple proteins) were associated with longitudinal health outcomes that are considered putative phenotypic metrics of aging (multimorbidity and death). We explore one molecular mechanisms of age-associated differences in protein levels by using methylation data. In addition, also leveraged publicly available genetic data to explore further the relationship between age-associated protein and aging disease. We have summarized this in the first paragraph of the Discussion.

“A proteomic analysis of chronological age was conducted to identify individuals who are experiencing accelerated health deterioration with aging based on their proteomic profile. The proteomic analysis identified 651 age-associated plasma proteins, many of which had a strong age-relationship independent of the presence of chronic diseases. Many of these proteins were previously described, 224 additional novel age-associated proteins are reported in the InCHIANTI study. Sex-stratified analysis showed that for most proteins, age-associations were similar in men and women. Proteins that systematically change with aging are also biomarkers of health longitudinally and risk biomarkers for mortality and accumulation of multi-morbidity over time in all subjects as well in persons that appear to be healthy by being free of any chronic diseases at the time of the evaluation. Analysis of DNA methylation data showed that methylation is one of the molecular mechanisms of age-associated differences in protein expression. Exploration of publicly available genetic data to assess the genetic relationship between age-associated protein and aging disease supported the clinical importance of these proteins. Finally, age acceleration measure using proteomic signature of age, which can be interpreted as a proxy measure for biological age, is associated with comorbidity and predicts future rate of change in multi-morbidity as well as all-cause mortality over a 18-year follow-up period.”

[Editors' note: further revisions were suggested prior to acceptance, as described below.]

Revisions:1) It seems the authors simply replaced those various labels with "B-H FDR". Hence, some values in the columns "B-H FDR" cannot be correct. For example, the values of "B-H FDR" are identical to "p" in Supplementary file 1B.

Thank you for catching the errors. As per the prior request by one of the reviewers, we changed all the headers of FDR corrected p-values to B-H FDR. However, in the case of Supplementary file 1B, there was an error where the FDR p-values were selected for both p-value and q-value columns. This has been corrected. The first 5 rows for age 41 from Supplementary file 1B.

2) No violation of the proportional hazards assumption should be clearly stated in the manuscript. As only the global p-value was given in the response, it is not clear whether the authors had checked the significance of every variable in their Cox model or not.

We had not checked the proportionality assumption for each model. However, we agree with the reviewer that this is an important step in ensuring the validity of the results. We included a test for proportionality for each model and included this information in Figure 4—source data 1. Of the 651 proteins tested, there was a violation for 26 proteins. For these proteins, the follow-up time interval was split into two (at the median) or three (1^st^, 3^rd^ quarter) based on the plot of the Schoenfeld residuals. Then a stratified analysis was conducted by including an interaction terms between time interval and proteins. Interestingly, the associations were only observed in the early time interval. These results are submitted as Figure 4—source data 2 and described in the text. The details are outlines in the Materials and methods, and the results changed to reflect the changes as shown below:

“The association of age-associated proteins with all-cause mortality was tested with cox proportional hazards model. Of the 651 age-associated proteins in the InCHIANTI study, the model for 26 proteins violated the proportional hazards assumption. Of the remaining 625 proteins, 497 proteins were associated with all-cause mortality and 193 of them remained significantly associated with mortality after adjusting for covariates (age, sex, study site) (Figure 4—source data 1). For the 26 proteins where the proportional hazard assumption was not met, stratified analyses showed that 25 of the proteins were predictive of all-cause mortality in the first time interval of 12 or 15 years (Figure 4—source data 2).”

“Association of protein with all-cause mortality was analyzed using follow-up time to death as the time variable. Cox proportional hazard models were assessed adjusting for covariates (age, sex, and study site) using the coxph function from the survival package For each fully adjusted model, the proportional hazards assumption was tested by examining the correlation of scaled Schoenfeld residuals against time using the cox.zph function. A global p-value ≤ 0.05 was considered a violation of proportional hazard assumption (26 proteins). For these proteins, a stratified analysis was conducted by splitting the follow-up period into two or three intervals and including an interaction term between protein and time-interval strata in the model. Based on the plot of the Schoenfeld residuals versus transformed time, follow-up time was split into two intervals at the mean follow-up year (15 years) for 18 proteins, and into 3 intervals for at 12 and 18 years (1st and 3rd quarter) for the remaining 8 proteins.”

3) The text "did not affect the results" is too vague. It should be clearly stated with more explanation about the model and results.

We included a clearer statement about the changes in the model and results when non-linear effects were accounted for.

“ The association of PROaccel remained significant and the point estimate remained at an annual change of 0.03 after account for non-linear effects with the inclusion of quadratic and/or cubic term for age in the model.”

4) All genes should not be used as the background reference because those included the genes for the proteins the authors didn't measure at all. When all genes were used as the reference set, such enrichment analysis likely identifies the enrichment among the 1301 measured proteins, which were proteins in plasma, comparing to all proteins. This explains why the authors found strong enrichments of inflammatory pathways.

As requested, we have performed new pathway over-representation analysis using the 1301 protein SOMAscan panel as a background reference instead of the entire human proteome. We appreciate the reviewer’s request for this re-analysis as it has significantly impacted the pathways. The new analysis remains enriched with inflammation pathways, albeit reduced from the previous analysis, and has highlighted several new pathways involved in regulation of gene expression. We have revised the supplemental table (Supplementary file 1E), figures (Figures 4A and 5A) and text, including the Results and Materials and methods sections, to reflect the new changes, summarized below:

“Among the proteins that were associated with age, there was an enrichment for inflammatory pathways (TNF-activated receptor activity and chemokine receptor binding), regulation of gene expression (DNA methylation, meiosis, epigenetic regulation of gene expression), and extracellular matrix (activation of matrix metalloproteinases, basement membrane, extracellular matrix organization) (Figure 4A).”

“Morbidity-associated proteins were strongly enriched for inflammatory pathways (interleukin-7 signaling, cytokine receptor interaction, senescence-associated secretory phenotype), regulation of IGF transport and uptake by IGFBPs, regulation of gene expression (HDMs demethylate histones, meiosis, SIRT1 negatively regulates rRNA, histone modifications, DNA methylation), and amyloid fiber formation (Figure 5A).”

“Pathway enrichment, and visualization

Gene set over-representation analyses were performed using the ConsensusPathDB-human tool, release 34 (15.01.2019)(70). Curated pathways for enrichment analysis were referenced from the following databases: Gene Ontology, WikiPathways, and Kegg pathways, and Reactome. Pathways were required to have a minimum of three observed proteins and a p-value < 0.01. Gene ontology terms were restricted to levels 3-5. For morbidity-associated proteins (Figure 5A), all pathways with enrichment q-values < 0.05 were kept, and the top 25 pathways were plotted. For mortality- associated proteins (Figure 4A), all pathways with enrichment q-values < 0.25 were kept. One duplicate named pathway appearing from multiple source databases was omitted manually. For background reference, a list of all 1301 proteins measured in the SOMAscan assay were used. Tables containing the pathways, observed corresponding proteins, reference pathway annotations, statistics, and source databases are included as a Supplementary file (Supplementary file 1E).

Heatmaps and dotplots were generated in R with the “ggplot2” package (70) and by comparing with SASP profiles on SASP Atlas (21) (www.saspatlas.com). Color palettes in R were generated with the “RColorBrewer” package (72).”

Figure 4A was replaced with this dotplot visualization of the gene overrepresentation analysis.

Figure 5A was replaced with this dotplot visualization of the gene overrepresentation analysis.